# PARETO SET LEARNING FOR NEURAL MULTI-OBJECTIVE COMBINATORIAL OPTIMIZATION

**Xi Lin, Zhiyuan Yang, Qingfu Zhang**
Department of Computer Science, City University of Hong Kong
`xi.lin@my.cityu.edu.hk`

## ABSTRACT

Multiobjective combinatorial optimization (MOCO) problems can be found in many real-world applications. However, exactly solving these problems would be very challenging, particularly when they are NP-hard. Many handcrafted heuristic methods have been proposed to tackle different MOCO problems over the past decades. In this work, we generalize the idea of neural combinatorial optimization, and develop a learning-based approach to approximate the whole Pareto set for a given MOCO problem without further search procedure. We propose a single preference-conditioned model to directly generate approximate Pareto solutions for any trade-off preference, and design an efficient multiobjective reinforcement learning algorithm to train this model. Our proposed method can be treated as a learning-based extension for the widely-used decomposition-based multiobjective evolutionary algorithm (MOEA/D). It uses a single model to accommodate all the possible preferences, whereas other methods use a finite number of solutions to approximate the Pareto set. Experimental results show that our proposed method significantly outperforms some other methods on the multiobjective traveling salesman problem, multiobjective vehicle routing problem, and multiobjective knapsack problem in terms of solution quality, speed, and model efficiency.

## 1 INTRODUCTION

Many real-world applications can be modeled as multiobjective combinatorial optimization (MOCO) problems (Ehrgott & Gandibleux, 2000). Examples include the multiobjective traveling salesman problem (MOTSP) (Lust & Teghem, 2010a), the multiobjective vehicle routing problem (MOVRP) (Jozefowiez et al., 2008) and the multiobjective knapsack problem (MOKP) (Bazgan et al., 2009). These problems have multiple objectives to optimize, and no single solution can optimize all the objectives at the same time. Instead, there is a set of Pareto optimal solutions with different trade-offs among the objectives.

It is very challenging to find all the exact Pareto optimal solutions for a MOCO problem. Actually, finding one single Pareto optimal solution can be NP-hard for many problems (Ehrgott & Gandibleux, 2000), and the number of Pareto solutions could be exponentially large with regard to the problem size (Ehrgott, 2005; Herzel et al., 2021). The decision-maker's preference among different objectives is usually unknown in advance, making it very difficult to reduce the problem into a single-objective one. Over the past several decades, many methods have been developed to find an approximate Pareto set for different MOCO problems within a reasonable computational time. These methods often need carefully handcrafted and specialized heuristics for each problem. It can be very labor-intensive in practice.

In many real-world applications, practitioners need to solve many different instances for the same particular problem, where the instances can be easily obtained or generated (Bengio et al., 2020). It is desirable to learn the patterns behind these problem instances explicitly or implicitly to design efficient algorithms (Cappart et al., 2021a). Machine learning techniques can be naturally used for this purpose. Some learning-based methods have been recently proposed for solving single-objective combinatorial optimization problems (Bengio et al., 2020; Vesselinova et al., 2020; Mazyavkina et al., 2021; Cappart et al., 2021a). In this work, we extend the learning-based method to solve MOCO problems in a principled way as shown in Figure 1. Our main contributions include:

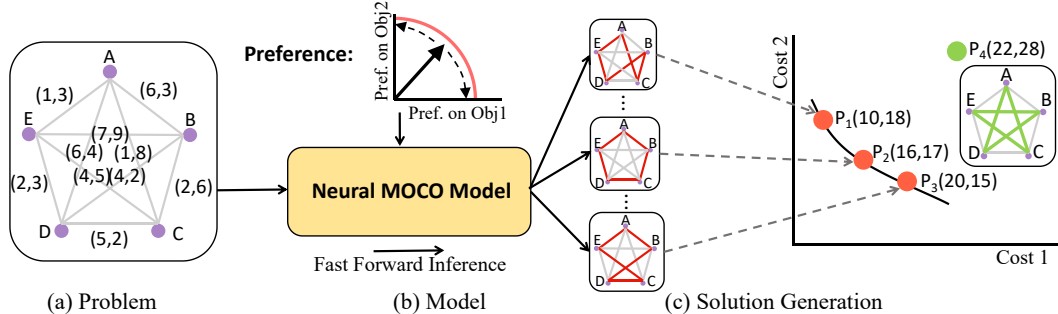

Figure 1: **Preference-Conditioned Neural Multiobjective Combinatorial Optimization: a)** The model takes a problem instance as its input. **b)** The decision makers assign their preferences on different objectives to the model. **c)** The model directly generates approximate Pareto solutions with different trade-offs via fast forward inference. In this example, the problem is the MOTSP with two cost objectives to minimize. The generated solutions $P_1$, $P_2$ and $P_3$ are different optimal trade-offs between the two cost objectives. The ideal model can generate solutions for all possible optimal trade-offs on the Pareto front and not generate a poor solution such as $P_4$.

- We propose a novel neural multiobjective combinatorial optimization method to approximate the whole Pareto set via a single preference-conditioned model. It allows decision makers to obtain any preferred trade-off solution without any search effort.

- We develop an efficient end-to-end reinforcement learning algorithm to train the single model for all different preferences simultaneously, and a simple yet powerful active adaption method to handle out-of-distribution problem instances.

- We conduct comprehensive experiments on MOTSP, MOVR and MOKP of different settings. The results show that our proposed method can successfully approximate the Pareto sets for different problems in an efficient way. It also significantly outperforms other methods in terms of solution quality, speed, and model efficiency.

## 2 BACKGROUND AND RELATED WORK

**Multiobjective Combinatorial Optimization (MOCO).** MOCO has been attracting growing research efforts from different communities over the past several decades (Sawaragi et al., 1985; Wallenius et al., 2008; Herzel et al., 2021). There are two main approaches to tackle the MOCO problems: the exact methods and the approximation methods (Ehrgott, 2005). Exact methods could be prohibitively costly when, as it often happens, the MOCO problem is NP-hard and the problem size is very large (Florios & Mavrotas, 2014). For this reason, many heuristics (Jaszkiewicz, 2002; Zhang & Li, 2007; Ehrgott & Gandibleux, 2008) and approximation methods (Papadimitriou & Yannakakis, 2000; Herzel et al., 2021) have been developed to find a manageable number of approximated Pareto solutions with a reasonable computational budget. However, these methods usually depend on carefully handcrafted designs for each specific problem (Ehrgott & Gandibleux, 2000), and the required effort is often nontrivial in real-world applications.

**Machine Learning for Combinatorial Optimization.** As summarized in Bengio et al. (2020), there are three main learning-based approaches for combinatorial optimization: learning to configure algorithms (Kruber et al., 2017; Bonami et al., 2018), learning alongside the algorithms (Lodi & Zarpellon, 2017; Gasse et al., 2019; Chen & Tian, 2019), and learning to directly predict the solutions (Nowak et al., 2018; Emami & Ranka, 2018; Larsen et al., 2018). Neural combinatorial optimization (NCO) belongs to the last category where the model directly produces a good solution for a given problem instance. Vinyals et al. (2015) proposed a pointer network to sequentially construct a solution for the TSP problem. Bello et al. (2017) made a critical improvement to use reinforcement learning to train the model, eliminating the impractical optimal solutions collection for NP-hard problems. Some other improvements on model structure and training procedure have been proposed in the past few years (Nazari et al., 2018; Deudon et al., 2018; Kool et al., 2019; Veličković & Blundell, 2021), especially with graph neural networks (GNNs) (Dai et al., 2017; Li et al., 2018; Joshi et al., 2019; Dwivedi et al., 2020; Drori et al., 2020). Recent efforts have been made on more

efficient learning strategies (Kwon et al., 2020; Karalias & Loukas, 2020; Lisicki et al., 2020; Geisler et al., 2022), learning-based graph search (Cappart et al., 2021b; Kool et al., 2021; Fu et al., 2021; Xin et al., 2021; Hudson et al., 2022), and iterative improvement methods (Wu et al., 2021; Ma et al., 2021; Li et al., 2021).

**Neural MOCO.** Most of the existing learning-based methods are for single-objective combinatorial problems. Recently, a few attempts have been made to solve MOCO problems (Li et al., 2020; Wu et al., 2020; Zhang et al., 2021a;b). These methods adopt the MOEA/D framework (Zhang & Li, 2007) to decompose a MOCO problem into a number of single-objective subproblems, and then build a set of models to solve each subproblem separately. However, since the number of Pareto solutions would be exponentially large (Ehrgott, 2005), the required number of models would be huge for finding the whole Pareto set. In this work, we propose a single preference-conditioned model for solving MOCO problems, with which the decision makers can easily obtain any trade-off solutions. The proposed single neural MOCO solver could be much easier to use in a real-world system (Veličković & Blundell, 2021), than those using a large set of different models.

# 3 PROBLEM FORMULATION

## 3.1 MULTIOBJECTIVE COMBINATORIAL OPTIMIZATION

A multiobjective combinatorial optimization (MOCO) problem can be defined as follows:

$$\min_{x \in \mathcal{X}} F(x) = (f_1(x), f_2(x), \ldots, f_m(x)), \tag{1}$$

where $\mathcal{X}$ is a discrete search space, and $F(x) = (f_1(x), \ldots, f_m(x))$ is an $m$-objective vector. Since the individual objectives conflict each other, no single solution can optimize all of them at the same time. Therefore, practitioners are interested in Pareto optimal solutions, defined as follows.

**Definition 1 (Pareto Dominance).** Let $x_a, x_b \in \mathcal{X}$, $x_a$ is said to dominate $x_b$ ($x_a \prec x_b$) if and only if $f_i(x_a) \leq f_i(x_b), \forall i \in \{1, ..., m\}$ and $f_j(x_a) < f_j(x_b), \exists j \in \{1, ..., m\}$.

**Definition 2 (Pareto Optimality).** A solution $x^* \in \mathcal{X}$ is a Pareto optimal solution if there does not exist $\hat{x} \in \mathcal{X}$ such that $\hat{x} \prec x^*$. The set of all Pareto optimal solutions is called the Pareto set, and the image of the Pareto set in the objective space is called the Pareto front.

Each Pareto solution represents an optimal trade-off among the objectives, and it is impossible to further improve one of the objectives without deteriorating any other objectives.

## 3.2 DECOMPOSITION AND PREFERENCE-BASED SCALARIZATION

Decomposition is a mainstream strategy for solving multiobjective optimization problem (Zhang & Li, 2007). It decomposes a multiobjective problem into a number of subproblems, each of which can be a single objective or multiobjective optimization problem. MOEA/D (Zhang & Li, 2007) and its variants (Trivedi et al., 2016) solve these subproblems in a collaborative manner and generate a finite set of Pareto solutions to approximate the Pareto front. The most widely used way for constructing a single objective subproblem is the preference-based scalarization (Ehrgott, 2005; Miettinen, 2012). For an $m$-objective optimization problem, a preference vector for the objective functions can be defined as $\lambda \in R^m$ that satisfies $\lambda_i \geq 0$ and $\sum_{i=1}^{m} \lambda_i = 1$.

**Weighted-Sum Aggregation** is the simplest approach. It defines the aggregation function to minimize in the subproblem associated with $\lambda$ as

$$g_{\text{ws}}(x|\lambda) = \sum_{i=1}^{m} \lambda_i f_i(x). \tag{2}$$

However, this approach can only find solutions on the convex hull of the Pareto front (Ehrgott, 2005).

**Weighted-Tchebycheff (Weighted-TCH) Aggregation** is an alternative approach to minimize:

$$g_{\text{tch}}(x|\lambda) = \max_{1 \leq i \leq m} \{\lambda_i | f_i(x) - z_i^* |\}, \tag{3}$$

where $z_i^* < \min_{x \in \mathcal{X}} f_i(x)$ is an ideal value for $f_i(x)$. Any Pareto optimal solution could be an optimal solution of problem (3) with a specific (but unknown) preference $\lambda$ (Choo & Atkins, 1983).

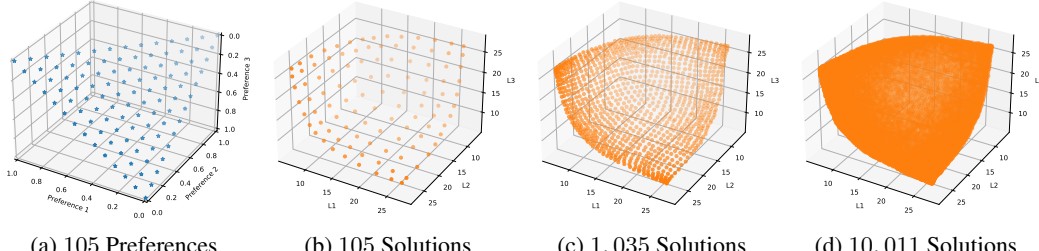

| (a) 105 Preferences | (b) 105 Solutions | (c) 1, 035 Solutions | (d) 10, 011 Solutions |

Figure 2: **Pareto Set Approximation for a New Problem Instance:** Our method with different numbers of preferences on three objective MOTSP with 50 nodes. **a)&b)** With a small number of preferences, our model can generate a sparse approximation to the Pareto set. **c)&d)** With a large number of preferences, it can generate a dense approximation. Our method can generate any trade-off solutions with a single model without searching, whereas other methods need to solve or build a model for each preference separately. More discussions can be found in Appendix D.3 D.5 D.6 D.7.

### 3.3 CURRENT DRAWBACKS AND OUR METHOD

**Drawbacks of Existing Methods.** For many MOCO problems, the size of the Pareto set would be exponentially large with respect to the input size (e.g., nodes in MOTSP). It is computationally impractical for existing methods to find the whole Pareto set (Herzel et al., 2021). For this reason, all of the existing heuristic-based and learning-based methods are to find a small subset of approximate Pareto solutions. Decision makers can only select solutions from this small set, which often does not contain their preferred solutions. In addition, scalarization may also produce a complicated single objective subproblem. For example, the Tchebycheff scalarized subproblem of MOTSP is not a classic TSP, and thus cannot be solved by the highly specialized TSP solvers such as LKH (Helsgaun, 2000) or Concorde (Applegate et al., 2007).

**Our Method.** Instead of finding a set of finite solutions, we propose a novel way to approximate the whole Pareto set using a single model. With our proposed model, decision makers can easily obtain any solution from the approximate Pareto set to satisfy their preferred trade-offs in real time as shown in Figure 2. This is a clear advantage to support interactive decision making. In addition, our proposed reinforcement learning based method can use a scalarization method to combine multiobjective rewards, and does not need to consider the problem-specific condition. In this paper, we mainly consider learning the whole Pareto front. It is possible to incorporate decision-maker's preferences on specific regions for model building and inference as discussed in Appendix D.6. We believe our proposed method is a new principled way to solve multiobjective combinatorial optimization problems.

## 4 THE PROPOSED MODEL: PREFERENCE-CONDITIONED NEURAL MOCO

### 4.1 PREFERENCE-CONDITIONED SOLUTION CONSTRUCTION

Decomposition and scalarization link preferences to their corresponding Pareto solutions. This work builds a preference-conditioned model to accommodate all the preferences. We use the MOTSP as an example to explain our model design. In an MOTSP instance $s$, a fully connected graph of $n$ nodes (cities) with $m$ distance metrics on each edge is given. A feasible solution is a tour that visits each city exactly once and returns to the starting city. The $i$-th objective to minimize is the tour length (total cost) based on the $i$-th distance metric. A tour can be represented as $\boldsymbol{\pi} = (\pi_1, \cdots, \pi_t, \cdots, \pi_n), \pi_t \in \{1, \cdots, n\}$, a permutation of all the nodes defining the order in which $n$ cities is visited. Our model defines a preference-conditioned stochastic policy $p_{\boldsymbol{\theta}(\lambda)}(\boldsymbol{\pi}|s)$ parameterized by $\boldsymbol{\theta}(\lambda)$ to construct a valid solution in sequence:

$$p_{\boldsymbol{\theta}(\lambda)}(\boldsymbol{\pi}|s) = \prod_{t=1}^{n} p_{\boldsymbol{\theta}(\lambda)}(\boldsymbol{\pi}_t|s, \boldsymbol{\pi}_{1:t-1}). \tag{4}$$

The goal is to learn an optimal preference-conditioned policy $p_{\boldsymbol{\theta}(\lambda)}(\pi|s)$ to construct tours with the lowest scalarized costs for each preference $\lambda$.

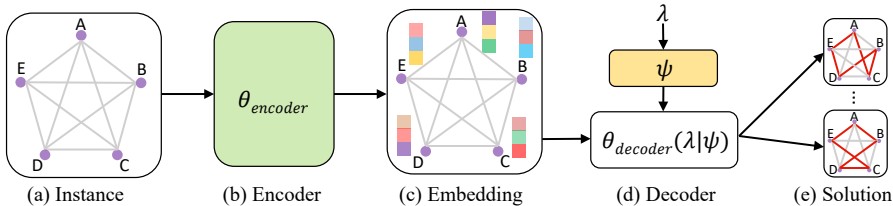

|     |     |     |     |     |
| --- | --- | --- | --- | --- |
| (a) Instance | (b) Encoder | (c) Embedding | (d) Decoder | (e) Solution |

Figure 3: **Preference-conditioned Neural MOCO Model: a)** The input is a problem instance $s$ (e.g., a graph). **b)** A shared attention encoder transfers the instance $s$ into a set of embeddings. **c)** The embeddings for all nodes would be used by the decoder multiple times with different preferences. **d)** A MLP takes the preference $\lambda$ as input, and generates the parameters for the decoder. **e)** The preference-conditioned attention decoder directly generates different approximate Pareto solutions for different preferences. The trainable parameters are in the **attention encoder** and **MLP model**.

## 4.2 THE PROPOSED MODEL

We propose to use an Attention Model (AM) (Kool et al., 2019) as our basic encoder-decoder model as shown in Figure 3. For the MOCO problems considered in this work, a preference-agnostic encoder is capable to transfer problem instances into embeddings (e.g., embedding for all cities) used in the preference-conditioned decoder. In our model, only the decoder's parameters $\boldsymbol{\theta}_{\text{decoder}}(\lambda)$ are conditioned on the preference $\lambda$:

$$\boldsymbol{\theta}(\lambda) = [\boldsymbol{\theta}_{\text{encoder}}, \boldsymbol{\theta}_{\text{decoder}}(\lambda)]. \tag{5}$$

**Preference-agnostic Encoder.** The encoder takes a problem instance $s$ (e.g., an MOTSP instance with $n$ cities) as its input, and outputs a set of $d$-dimensional node embeddings $\{\boldsymbol{h}_1, \cdots, \boldsymbol{h}_n\}$ for each city. For a given instance, the same embeddings can be used for different preferences. Hence we only need a single forward pass for the dense encoder. We use the attention-based encoder as in Kool et al. (2019) for all preferences.

**Preference-based Attention Decoder.** The decoder has the same structure as in the attention-based model (Kool et al., 2019), but with parameters $\boldsymbol{\theta}_{\text{decoder}}(\lambda) = [W_Q(\lambda), W_K(\lambda), W_V(\lambda), W_{\text{MHA}}(\lambda)]$ conditioned on the preference $\lambda$. It takes the nodes embeddings for all cities as input, and sequentially selects the next node $\boldsymbol{\pi}_t$ with probability $p_{\boldsymbol{\theta}(\lambda)}(\boldsymbol{\pi}_t|s, \boldsymbol{\pi}_{1:t-1})$.

At time step $t$, the decoder first constructs a context embedding $\hat{\boldsymbol{h}}_{(C)} = [\boldsymbol{h}_{\boldsymbol{\pi}_1}, \boldsymbol{h}_{\boldsymbol{\pi}_{t-1}}]W_Q(\lambda)$ from the first selected node $\boldsymbol{h}_{\boldsymbol{\pi}_1}$, and the last selected node $\boldsymbol{h}_{\boldsymbol{\pi}_{t-1}}$. The matrix $W_Q(\lambda) \in \mathbb{R}^{2d \times d}$ projects the concatenated $2d$-dimensional vector to a $d$-dimensional vector. Then we further aggregate the context embedding via a Multi-Head Attention (MHA) (Vaswani et al., 2017) with the embeddings for all cities $\{\boldsymbol{h}_1, \cdots, \boldsymbol{h}_n\}$:

$$\boldsymbol{h}_{(C)} = \textbf{MHA}(Q = \hat{\boldsymbol{h}}_{(C)}, K = \{\boldsymbol{h}_1, \cdots, \boldsymbol{h}_n\}W_K(\lambda), V = \{\boldsymbol{h}_1, \cdots, \boldsymbol{h}_n\}W_V(\lambda))W_{\textbf{MHA}}(\lambda), \quad (6)$$

where $Q, K, V$ are the query, key and value for MHA, respectively. $W_{\textbf{MHA}}(\lambda)$ represents the MHA parameters. The context embedding $h_{(C)}$ contains all information for the instance and the current partial tour at step $t$. We can calculate the logit for selecting each city with its embedding $h_j$:

$$\text{logit}_j = \begin{cases} C \cdot \tanh(\frac{\boldsymbol{h}_{(C)}^T \boldsymbol{h}_j}{\sqrt{d}}) & \text{if } j \neq \boldsymbol{\pi}_{t'} \quad \forall t' < t, \\ -\infty & \text{otherwise.} \end{cases} \tag{7}$$

All already visited cities are masked with $-\infty$ and will not be selected as the next city. The logits of the rest cities are clipped into $[-C, C]$ ($C = 10$) as in the AM model (Kool et al., 2019). The probability for choosing the $j$-th city at time step $t$ can be calculated as $p_{\boldsymbol{\theta}(\lambda)}(\boldsymbol{\pi}_t = j|s, \boldsymbol{\pi}_{1:t-1}) = e^{\text{logit}_j}/\sum_k e^{\text{logit}_k}$. With this probability, the decoder can construct a feasible tour.

One remaining designing issue is how to generate the preference-conditioned parameters $\boldsymbol{\theta}_{\text{decoder}}(\lambda)$. Multiplicative interactions (Jayakumar et al., 2020) and hypernetwork (Schmidhuber, 1992; Ha et al., 2017) provide a powerful and efficient way for conditional computation, which is widely used for transfer learning (von Oswald et al., 2020; Ehret et al., 2021; Lin et al., 2020; Navon et al., 2021). We use a simple MLP hypernetwork $\boldsymbol{\theta}_{\text{decoder}}(\lambda) = \text{MLP}(\lambda|\boldsymbol{\psi})$ to generate the decoder parameters conditioned on the preference. The details of our proposed model can be found in Appendix B.

---

**Algorithm 1** Neural MOCO Training

---

1: **Input:** preference distribution $\Lambda$, instances distribution $\mathcal{S}$, number of training steps $T$, number of preferences per iteration $K$, batch size $B$, number of tours $N$
2: Initialize the model parameters $\boldsymbol{\theta}$
3: **for** $t = 1$ to $T$ **do**
4: $\quad \lambda_k \sim \textbf{SamplePreference}(\Lambda) \quad \forall k \in \{1, \cdots, K\}$
5: $\quad s_i \sim \textbf{SampleInstance}(\mathcal{S}) \quad \forall i \in \{1, \cdots, B\}$
6: $\quad \boldsymbol{\pi}_{ki}^j \sim \textbf{SampleTour}(p_{\boldsymbol{\theta}(\lambda_k)}(\cdot|s_i)) \quad \forall k, i \quad \forall j \in \{1, \cdots, N\}$
7: $\quad b(s_i|\lambda_k) \leftarrow \frac{1}{N} \sum_{j=1}^N L(\boldsymbol{\pi}_{ki}^j|\lambda_k, s_i) \quad \forall k \in \{1, \cdots, K\} \quad \forall i \in \{1, \cdots, B\}$
8: $\quad \nabla \mathcal{J}(\boldsymbol{\theta}) \leftarrow \frac{1}{KBN} \sum_{k=1}^K \sum_{i=1}^B \sum_{j=1}^N [(L(\boldsymbol{\pi}_{ki}^j|\lambda_k, s_i) - b(s_i|\lambda_k))\nabla_{\boldsymbol{\theta}(\lambda_k)} \log p_{\boldsymbol{\theta}(\lambda_k)}(\boldsymbol{\pi}_{ki}^j|s_i)]$
9: $\quad \theta \leftarrow \textbf{ADAM}(\theta, \nabla \mathcal{J}(\boldsymbol{\theta}))$
10: **end for**
11: **Output:** The model parameter $\boldsymbol{\theta}$

---

**Instance Augmentation for MOCO.** Our proposed model only has a small extra computational and memory overhead to the original single-objective AM solver. We keep our model as simple as possible, making it easy for our approach to use other models and other improvements developed for single-objective NCO. These properties are crucially important for generalizing the NCO to multiobjective problems. In this work, we simply extend the instance augmentation method (Kwon et al., 2020) to MOCO. The details can be found in Appendix B.1.

## 5 PREFERENCE-CONDITIONED MULTIOBJECTIVE POLICY OPTIMIZATION

### 5.1 COST FUNCTION

Our proposed node selection strategy guarantees that the model can always generate feasible solutions. In this section, we develop an efficient multiobjective policy optimization method to train the model for all the preferences simultaneously. For an MOTSP problem, the objective functions are a vector of $m$ different costs (i.e. lengths) for a tour $L(\boldsymbol{\pi}) = [L_1(\boldsymbol{\pi}), \cdots, L_m(\boldsymbol{\pi})]$. We can define a weighted-Tchebycheff scalarized cost for each preference $\lambda$:

$$L(\boldsymbol{\pi}|\lambda) = \max_{1 \leq i \leq m} \{\lambda_i |L_i(\boldsymbol{\pi}) - (z_i^* - \varepsilon)|\}, \tag{8}$$

where $z_i^*$ is an ideal cost for the $i$-th objective. For a given instance $s$, our goal is to minimize the expected cost for all preferences:

$$\mathcal{J}(\boldsymbol{\theta}|s) = \mathbb{E}_{\lambda \sim \Lambda, \boldsymbol{\pi} \sim p_{\boldsymbol{\theta}(\lambda)}(\cdot|s)} L(\boldsymbol{\pi}|\lambda), \tag{9}$$

where $\Lambda$ is the uniform distribution over all valid preferences. To train the model, we repeatedly sample different instances $s \sim \mathcal{S}$ at each iteration. We define the training loss as $\mathcal{J}(\boldsymbol{\theta}) = \mathbb{E}_{s \sim \mathcal{S}} \mathcal{J}(\boldsymbol{\theta}|s)$.

### 5.2 MULTIOBJECTIVE REINFORCE

For a given instance $s$ and a specific preference $\lambda$, we use the REINFORCE (Williams, 1992) to estimate the gradient for the preference-conditioned scalar cost:

$$\nabla \mathcal{J}(\boldsymbol{\theta}|\lambda, s) = \mathbb{E}_{\boldsymbol{\pi} \sim p_{\boldsymbol{\theta}(\lambda)}(\cdot|s)}[(L(\boldsymbol{\pi}|\lambda, s) - b(s|\lambda))\nabla_{\boldsymbol{\theta}(\lambda)} \log p_{\boldsymbol{\theta}(\lambda)}(\boldsymbol{\pi}|s)], \tag{10}$$

where $b(s|\lambda)$ is the baseline of expected cost to reduce the gradient variance. This gradient can be estimated by Monte Carlo sampling. At each update step, we randomly sample $K$ preference $\{\lambda_1, \cdots, \lambda_K\} \sim \Lambda$, $B$ instances $\{s_1, \cdots, s_B\} \sim \mathcal{S}$, and $N$ different tour $\{\boldsymbol{\pi}_i^1, \cdots, \boldsymbol{\pi}_i^N\} \sim p_{\boldsymbol{\theta}(\lambda_k)}(\cdot|s_i)$ for each $\lambda_k$-$s_i$ combination. The approximated gradient is:

$$\nabla \mathcal{J}(\boldsymbol{\theta}) \approx \frac{1}{KBN} \sum_{k=1}^K \sum_{i=1}^B \sum_{j=1}^N [(L(\boldsymbol{\pi}_i^j|\lambda_k, s_i) - b(s_i|\lambda_k))\nabla_{\boldsymbol{\theta}(\lambda_k)} \log p_{\boldsymbol{\theta}(\lambda_k)}(\boldsymbol{\pi}_i^j|s_i)]. \tag{11}$$

We use the shared baseline $b_{\text{shared}}(s_i|\lambda_k) = \frac{1}{N} \sum_{j=1}^N L(\boldsymbol{\pi}_{ki}^j|\lambda_k, s_i)$ over $N$ sampled tours for each $\lambda_k - s_i$ combination. The starting node for each tour $\boldsymbol{\pi}_{ki}^j$ is chosen in random to force diverse rollouts as proposed in (Kwon et al., 2020). The algorithm is shown in **Algorithm 1**.

## 5.3 ACTIVE ADAPTION

We also propose a simple yet powerful active adaption approach to further adjust the whole model to approximate the Pareto front for a given test instance in Appendix B.3. The proposed method does not depend on specific instance distribution $\mathcal{S}$, and is suitable for out-of-distribution adaption.

## 6 EXPERIMENTS

**Problems and Model Setting.** We consider MOTSP (Lust & Teghem, 2010a), MOCVRP (Lacomme et al., 2006) and MOKP (Bazgan et al., 2009) in our experimental studies, and use the same model settings for all problems with different task-specific input sizes and mask methods. The main policy model encoder is the Attention Model (Kool et al., 2019) and the hypernetwork is an MLP. We randomly generate $100,000$ problem instances on the fly for each epoch, and train the model for 200 epochs. The optimizer is ADAM with learning rate $\eta = 10^{-4}$ and weight decay $10^{-6}$. We train our models on a single RTX 2080-Ti GPU, and it costs about 10 minutes for an epoch on MOTSP100. We give detailed model settings, problem formulations, and more experimental results in Appendix BCD. The source code can be found in `https://github.com/Xi-L/PMOCO`.

**Baseline.** We call our proposed preference-conditioned multiobjective combinatorial optimization as **P-MOCO**. We compare it with three widely-used evolutionary algorithm frameworks for MOCO: **MOGLS** (Jaszkiewicz, 2002) is a multiobjective genetic local search algorithm, **NSGAII** (Deb et al., 2002) is a Pareto dominance-based multiobjective genetic algorithm, and **MOEA/D** (Zhang & Li, 2007) is a decomposition-based multiobjective evolutionary algorithm. All these algorithm frameworks need problem-specific heuristics to generate and search feasible solutions for different problems. We also compare **P-MOCO** with two other learning-based methods: **DRL-MOA** (Li et al., 2020) decomposes a MOCO with different preferences and builds a Pointer Network (Vinyals et al., 2015; Bello et al., 2017) to solve each subproblem, and **AM-MOCO** is a multi-models variant of our proposed model, which builds Attention Model (Kool et al., 2019) for each subproblem. The **Weight-Sum** scalarization of MOTSP and MOKP are their respective single-objective counterpart. Therefore, we also compare our method with the approach that uses some state-of-the-art single-objective solvers for each weight-sum subproblem.

Table 1: Model Information for the learning-based methods.

|  | Model | #Models | #Params | #Pref. |
|---|---|---|---|---|
| DRL-MOA | Pointer-Network | 101 | $101\times 0.2\text{M} = 20.2\text{M}$ (7%) | Fixed 101 |
| AM-MOCO | Attention Model | 101 | $101\times 1.3\text{M} = 131.3\text{M}$ (1.1%) | Fixed 101 |
| P-MOCO (Ours) | Pref-Conditioned AM | 1 | 1.4M | Flexible |

**Model Information** for the learning-based methods is shown in Table 1. Our model supports flexible preference assignment and only has $1.1\%$ total parameters to the multi-model counterpart.

**Inference and Metrics.** We report the results and run time for solving 200 random test instances for each problem, with normally 101 to 105 different trade-offed solutions, and up to $10,011$ solutions for our proposed method. In most cases, we report our model's zero-shot generalization performance without any search and fine-tune. We use the hypervolume indicator (Zitzler et al., 2003) to measure the performance for each method. For a set $P \subset \mathbb{R}^m$ in the objective space, we can find a reference point $r^*$ that dominated by all solutions in $P$, and define the hypervolume HV($P$) as volume for:

$$S = \{r \in \mathbb{R}^m \mid \exists y \in P \text{ such that } y \prec r \prec r^*\}, \tag{12}$$

where $\text{HV}(P) = \textbf{Vol}(S)$. In general, the larger the hypervolume, the better the solution set tends to be. The ground truth Pareto set always has the largest hypervolume. We report the normalized hypervolume values in $[0, 1]$ with respect to the same $r^*$ for all the methods, and also the ratios of hypervolume difference to our method. A Wilcoxon rank-sum test with a significance level $1\%$ is conducted to compare the results for each experiment. More details can be found in Appendix D.1.

### 6.1 RESULTS AND ANALYSIS

**MOTSP.** The results on two and three objective MOTSP are shown in Table 2 and Table 3 respectively. MOGLS, NSGAII and MOEA/D all use 2-opt local search heuristic (Jaszkiewicz, 2002) to

Table 2: Experimental results on two-objective MOTSP, MOCVRP and MOKP with different input sizes (20,50 and 100). HV is the hypervolume metric, Gap is the ratio of hypervolume difference with respect to our method, and Time is the running run for solving 200 random test instances. The best result and its statistically indifferent results are highlighted in **bold**.

| MOTSP | | | | | | | | | |
|---|---|---|---|---|---|---|---|---|---|
| | MOTSP20 | | | MOTSP50 | | | MOTSP100 | | |
| Method | HV | Gap | Time | HV | Gap | Time | HV | Gap | Time |
| Weight-Sum LKH | 0.51 | 2.44% | (3.9m) | **0.58** | **-0.26%** | (42m) | **0.68** | **-0.58%** | (3.1h) |
| Weight-Sum OR Tools | 0.48 | 7.29% | (3.7m) | 0.52 | 9.66% | (38m) | 0.63 | 5.64% | (2.8h) |
| MOGLS-TSP | 0.48 | 7.96% | (1.7h) | 0.51 | 10.7% | (4.5h) | 0.59 | 14.7% | (12h) |
| NSGAII-TSP | 0.47 | 9.88% | (45m) | 0.46 | 19.7% | (49m) | 0.51 | 26.5% | (53m) |
| MOEA/D-TSP | 0.48 | 7.82% | (3.1h) | 0.47 | 18.9% | (3.3h) | 0.52 | 25.0% | (3.7h) |
| DRL-MOA (101 models) | 0.46 | 11.3% | (2s) | 0.49 | 15.0% | (5s) | 0.57 | 17.0% | (15s) |
| AM-MOCO (101 models) | **0.52** | **0.22%** | (2s) | 0.57 | 1.72% | (7s) | 0.67 | 0.72% | (18s) |
| P-MOCO (101 pref.) | **0.52** | **0.19%** | (2s) | 0.57 | 1.80% | (5s) | 0.67 | 0.60% | (16s) |
| P-MOCO (101 pref., aug) | **0.52** | **0.00%** | (2.1m) | 0.58 | 0.00% | (4.2m) | 0.67 | 0.00% | (12m) |
| MOCVRP | | | | | | | | | |
| | MOCVRP20 | | | MOCVRP50 | | | MOCVRP100 | | |
| Method | HV | Gap | Time | HV | Gap | Time | HV | Gap | Time |
| MOGLS-MOCVRP | 0.33 | 6.66% | (2.9h) | 0.39 | 10.5% | (4.1h) | 0.38 | 13.0% | (5h) |
| NSGAII-MOCVRP | 0.31 | 12.3% | (39m) | 0.36 | 18.0% | (39m) | 0.35 | 20.8% | (41m) |
| MOEAD-MOCVRP | 0.29 | 17.9% | (1.7h) | 0.33 | 24.8% | (1.8h) | 0.34 | 22.3% | (2h) |
| DRL-MOA (101 models) | 0.32 | 9.19% | (6s) | 0.37 | 16.0% | (15s) | 0.37 | 16.0% | (33s) |
| AM-MOCO (101 models) | 0.35 | 0.71% | (8s) | 0.43 | 2.84% | (21s) | 0.43 | 3.04% | (39s) |
| P-MOCO (101 pref.) | 0.35 | 0.58% | (8s) | 0.44 | 0.65% | (18s) | 0.44 | 0.51% | (36s) |
| P-MOCO (101 pref., aug) | **0.35** | **0.00%** | (1m) | **0.44** | **0.00%** | (2.1m) | **0.45** | **0.00%** | (4.5m) |
| MOKP | | | | | | | | | |
| | MOKP50 | | | MOKP100 | | | MOKP200 | | |
| Method | HV | Gap | Time | HV | Gap | Time | HV | Gap | Time |
| Weight-Sum DP | 0.70 | 2.11% | (16m) | 0.83 | 2.74% | (1.5h) | 0.66 | 2.86% | (4.2h) |
| Weight-Sum Greedy | 0.62 | 14.0% | (2s) | 0.77 | 10.1% | (4s) | 0.63 | 7.49% | (7s) |
| MOGLS-KP | 0.58 | 19.1% | (1.8h) | 0.72 | 15.6% | (3.9h) | 0.62 | 9.49% | (9.8h) |
| NSGAII-KP | 0.56 | 22.0% | (34m) | 0.69 | 19.8% | (33m) | 0.58 | 14.7% | (34m) |
| MOEAD-KP | 0.59 | 18.2% | (1.5h) | 0.76 | 11.1% | (1.4h) | 0.65 | 5.38% | (1.6h) |
| DRL-MOA (101 models) | 0.63 | 12.9% | (5s) | 0.78 | 9.32% | (17s) | 0.65 | 4.39% | (35s) |
| AM-MOCO (101 models) | **0.72** | **0.21%** | (7s) | **0.86** | **0.41%** | (16s) | **0.68** | **0.32%** | (42s) |
| P-MOCO (101 pref.) | **0.72** | **0.00%** | (7s) | **0.86** | **0.00%** | (15s) | **0.68** | **0.00%** | (40s) |

search for promising solutions. We also include two weight-sum scalarization baselines with the state-of-the-art LKH solver (Helsgaun, 2000; Tinós et al., 2018) and Google OR tools (Perron & Furnon, 2019). For the bi-objective problems, our proposed method with a single model has similar performances compared with AM-MOCO on all problems. It achieves the best performance with instance augmentation, which significantly outperforms other methods but is beaten by the LKH solver. For the three objective problems, our method can further improve its performance by generating much more trade-off solutions within a reasonable amount of time, which other methods cannot do. As shown in Figure 2 and Figure 5, our method can successfully learn the mapping from preferences to the corresponding solutions, and can generate a good prediction to the whole Pareto front. Decision makers can easily obtain any preferred trade-off solutions as they like. This flexibility could be desirable in many real-world applications. More discussion on the connection between the preference and Pareto solution for three-objective TSP can be found in Appendix D.5 D.6 D.7.

**MOCVRP.** In this problem, each node has a demand, and we need to construct multiple return routes for a vehicle with a fixed capacity from the same depot to handle all demands. The objectives we consider are to minimize the tour length for all routes and also the tour length for the longest routes (the makespan in scheduling) (Lacomme et al., 2006). All the non-learning algorithm frameworks use the problem-specific constructive heuristics and local search method proposed in Lacomme et al. (2006) to search feasible non-dominated solutions. The results in Table 2 show that our method significantly outperforms the non-learning heuristics in terms of both solution quality

Table 3: Experimental results on three-objective MOTSP with different input sizes (20, 50 and 100). The best result and its statistically indifferent results are highlighted in **bold**.

| Method | MOTSP20 | | | MOTSP50 | | | MOTSP100 | | |
|---|---|---|---|---|---|---|---|---|---|
| | HV | Gap | Time | HV | Gap | Time | HV | Gap | Time |
| Weight-Sum LKH | 0.32 | 5.83% | (4.5m) | 0.36 | 5.48% | (44m) | 0.45 | 4.03% | (3.5h) |
| Weight-Sum OR Tools | 0.31 | 8.48% | (4.1m) | 0.31 | 15.5% | (39m) | 0.41 | 11.2% | (3.1h) |
| MOGLS-MOTSP | 0.27 | 19.6% | (2h) | 0.27 | 27.4% | (5h) | 0.33 | 30.5% | (13h) |
| NSGAII-MOTSP | 0.20 | 39.9% | (47m) | 0.18 | 49.3% | (56m) | 0.25 | 48.4% | (1h) |
| MOEA/D-MOTSP | 0.21 | 36.6% | (3.3h) | 0.20 | 46.7% | (3.7h) | 0.27 | 42.7% | (4.1h) |
| DRL-MOA (105 models) | 0.26 | 21.6% | (2s) | 0.28 | 23.0% | (5s) | 0.34 | 28.8% | (19s) |
| AM-MOCO (105 models) | 0.32 | 5.52% | (2s) | 0.33 | 6.02% | (7s) | 0.42 | 6.53% | (21s) |
| P-MOCO (105 pref.) | 0.32 | 5.65% | (2s) | 0.36 | 5.76% | (6s) | 0.44 | 5.52% | (19s) |
| P-MOCO (105 pref., aug) | 0.32 | 5.46% | (2.1m) | 0.36 | 5.43% | (6.5m) | 0.45 | 4.39% | (20m) |
| P-MOCO (1,035 pref.) | 0.33 | 2.08% | (19s) | 0.36 | 4.03% | (48s) | 0.46 | 2.12% | (2.9m) |
| P-MOCO (10,011 pref.) | **0.34** | **0.00%** | (3m) | **0.38** | **0.00%** | (9m) | **0.47** | **0.00%** | (33m) |

and running time. It also outperforms AM-MOCO with 100 individual models, which could be due to the asymmetric objective scales. We provide further analysis in Appendix D.4.

**MOKP.** The multiobjective 0-1 knapsack problem can be found in many real-world applications (Bazgan et al., 2009). We consider the uni-dimension problem, where each item has multiple values and one weight. The goal is to select a subset of items to maximize all obtained values with a weight constraint. The non-learning methods use binary coding with a greedy transformation heuristic to maintain feasibility (Ishibuchi et al., 2014). We also include weight-sum scalarization baselines with dynamic programming (DP) and a strong greedy search based on the value-weight ratio. According to the results in Table 2, our method has the best performance on all problems. The DP method is also outperformed by our method since the weight-sum scalarization can only find the convex hull of the Pareto front. The Tchebycheff scalarization of MOKP is not a KP problem, while our method is more flexible to use Tchebycheff scalarization on the reward function. We also report the results on 10 objective MOKP100 and the generalization performance to problem with 500 items in Appendix D.8.

**Out-of-Distribution Problems and Active Adaption.** We also validate the generalization performance of our method on 6 out-of-distribution (OOD) MOTSP problems from Fonseca et al. (2006). Their ground truth Pareto fronts can be obtained by exhaustive search. The results are shown in Appendix D.2 due to the page limit. With active adaption, our method can achieve good performance (1% - 1.5% HV gap to the ground truth Pareto fronts) on these OOD problems.

# 7 CONCLUSION AND FUTURE WORK

**Conclusion.** We have proposed a novel preference-conditioned method to approximate the whole Pareto front for MOCO problems using a single model. It allows decision makers to directly obtain any trade-off solutions without any search procedure. Experiments on different problems have shown that our proposed method significantly outperforms other methods in terms of performance, speed and model efficiency. We believe the proposed method is a principled way for solving MOCO.

**Future Work.** In a sense, our method can be regarded as a learning version of the decomposition-based algorithm (MOEA/D (Zhang & Li, 2007)) dealing with all the possible trade-off preferences. Instead of maintaining a set of finite solutions as in other MOEA/D vaiants (Trivedi et al., 2016), we build a single learning-based model to solve the subproblems for all the preferences simultaneously in a collaborative manner. We believe the single-model-for-all-preference approach is a promising alternative to the current default finite-population-based methods, and it could be an important research direction for multiobjective optimization. Our method can be further improved with other advanced models and efficient multiobjective training procedures. In the future, we will study fundamental issues of multiobjective optimization (e.g., convergence v.s. diversity, exploitation v.s. exploration trade-off) for Pareto set learning methods.

**Limitation.** It is very difficult to give a convergence guarantee for learning-based MOCO, where each preference-based subproblem could be already NP-hard, and the number of Pareto solutions is exponentially large with respect to the input size. See detailed discussion in Appendix A.

ACKNOWLEDGMENTS

We thank Prof. Hisao Ishibuchi for his valuable comments on an earlier version of this work. This work was supported by the Hong Kong General Research Fund (11208121, CityU-9043148).

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

We provide more discussion, details on the method and MOCO problems, experimental results and analysis in this appendix. Specifically:

- **Learning Ability and Approximation Analysis:** We discuss the learning ability of our proposed method, and give a thorough analysis on its approximation ability in Section A.

- **Method and Problem Details:** We provide the details on our proposed model and the MOCO problems in Section B and Section C, respectively.

- **More Experimental Results:** We give a detailed introduction to the hypervolume indicator, and present more experimental results with analysis in Section D.

## A  PARETO SET LEARNING AND APPROXIMATION ANALYSIS

### A.1  PARETO SET LEARNING AND CONVERGENCE GUARANTEE

In this work, we have proposed a novel neural combinatorial optimization (NCO) method to approximate the whole Pareto set for MOCO problems with a single model. The proposed learning-based MOCO solver can directly generate arbitrary trade-off solutions without extra optimization. We believe it is a principled way to solve MOCO problems.

However, the lack of an exact optimality guarantee is a limitation of the proposed method, which is also the case for previous work on single-objective neural combinatorial optimization (Vinyals et al., 2015; Bello et al., 2017; Kool et al., 2019). This limitation is mainly due to the fact that many single-objective combinatorial optimization (CO) problems are NP-hard, and the size of Pareto sets for a MOCO problem would be exponentially huge, which makes it very difficult to exactly solving the problems (Ehrgott, 2005; Herzel et al., 2021). In addition, the training for the parameterized policy (neural network model) cannot guarantee to fit all training problems perfectly. The generalization ability to problem instances with different patterns (out-of-distribution generalization) is another critical issue that makes it difficult to give an exact optimality guarantee to the proposed learning-based algorithm.

On the other hand, our proposed model is an efficient mapping from the preferences to the corresponding approximate set of the Pareto optimal solutions. It provides a flexible way for decision makers to obtain an approximate solution with their preferred trade-off directly. The experimental results also show that our proposed method can generate good approximate Pareto sets for three different MOCO problems. In the next subsection, we provide a thorough discussion on the approximation ability of our proposed method.

### A.2  APPROXIMATION ANALYSIS

For a MOCO problem, the number of Pareto solutions could be exponentially large with respect to its input size, which makes the problem intractable (Ehrgott, 2005; Herzel et al., 2021). The preference-based scalarization methods and decomposition methods (Choo & Atkins, 1983; Zhang & Li, 2007) we used provides a principled way to link the Pareto solutions with preference, allowing us to tackle the problem in a systematic manner. In this work, we propose to approximately solve the scalarized subproblem with all preferences via a single model.

We first briefly review the weighted scalarization method and its Pareto optimality guarantee as discussed in the main paper. Then we provide further discussion on the approximation analysis.

Our proposed method decomposes a MOCO problem into preference-based subproblems with the weighted-Tchebycheff scalarization (Weighted-TCH):

$$\min_{x \in \mathcal{X}} g_{\text{tch}}(x|\lambda) = \min_{x \in \mathcal{X}} \max_{1 \leq i \leq m} \{\lambda_i |f_i(x) - (z_i^* - \varepsilon)|\}, \tag{13}$$

where $z_i^*$ is the ideal value for objective $f_i(x)$ (e.g., the lower bound), and $u_i^* = z_i^* - \varepsilon$ is a utopia value with small positive component $\varepsilon$. The preference vector $\lambda \in \mathbb{R}^m$ satisfies $\lambda_i \geq 0$ and $\sum_{i=1}^m \lambda_i = 1$, where $\lambda_i$ is the preference for the $i$-th objective. This approach has a desirable property:

**Lemma 1 (Choo & Atkins (1983)).** *A feasible solution $x \in \mathcal{X}$ is Pareto optimal if and only if there is a weight vector $\lambda > 0$ such that $x$ is an optimal solution to the problem (13).*

According to **Lemma 1**, we can obtain any Pareto solution by solving the Weighted-TCH subproblem with a specific weight. However, the weight for each Pareto solution depends on its objective values, which are not known in advance (Sawaragi et al., 1985; Ehrgott, 2005). The decision-maker still needs to solve multiple subproblems with different preferences to find a desirable solution. To find the whole Pareto set, it needs to solve an exponentially huge number of subproblems.

Given a problem instance $s$, our proposed model provides a single mapping function $x_\lambda = h(\lambda)$ from any preference $\lambda$ to its corresponding solution $x_\lambda$, which is constructed by the preference-based policy $p_{\boldsymbol{\theta}(\lambda)}(x|s)$. In the ideal case, if all generated solutions $x_\lambda$ are the optimal solutions $x_\lambda^*$ of problem (13) with preference $\lambda$, according to **Lemma 1**, our proposed model can generate the whole Pareto set (all Pareto optimal solutions) for the original MOCO problem.

In practice, we are interested in the proposed method's approximation ability. We find that its performance strongly depends on the approximation ability of the parameterized policy (neural network model) on the single-objective scalarized subproblem. We first give an informal claim on our method's approximation ability, then provide detailed explanations and discussions.

**(Informal) Claim 1.** If the proposed method can approximately solve the subproblem (13) with any preference $\lambda$, it can generate a good approximation to the whole Pareto set for the MOCO problem.

To support this claim, we follow the traditional $\varepsilon$-Pareto approximate method for MOCO problems (Papadimitriou & Yannakakis, 2000; Herzel et al., 2021). First, an $\varepsilon$-Pareto domination relation between two individual solutions can be defined as:

**Definition 3 ($\varepsilon$-Pareto Domination).** For a MOCO problem and an $\varepsilon > 0$, let $x_a, x_b \in \mathcal{X}$, $x_a$ is said to $\varepsilon$-dominate $x_b$ ($x_a \prec_\varepsilon x_b$) if $f_i(x_a) \leq (1+\varepsilon)f_i(x_b), \forall i \in \{1, \cdots, m\}$.

This definition is a natural generalization from the $(1 + \varepsilon)$-approximation for single-objective optimization. With this concept, an $\varepsilon$-approximate Pareto set (Papadimitriou & Yannakakis, 2000) can be defined as:

**Definition 4 ($\varepsilon$-Approximate Pareto Set).** For an $\varepsilon > 0$, a set $\mathcal{P}_\varepsilon \subset \mathcal{X}$ is an $\varepsilon$-approximate Pareto set, if for any feasible solution $x \in \mathcal{X}$, there exists a solution $x' \in \mathcal{P}_\varepsilon$ such that $x' \prec_\varepsilon x$.

In other words, all feasible solutions of the MOCO problem can be almost dominated by some solutions in $\mathcal{P}_\varepsilon$ (Papadimitriou & Yannakakis, 2000). When the Pareto set is intractable and hard to find, the $\varepsilon$-approximate Pareto set would be a reasonable choice to achieve in practice. Each MOCO problem has a unique Pareto set, but can have different $\varepsilon$-approximate Pareto sets. The ability of our proposed method to find an $\varepsilon$-approximate Pareto set strongly depends on its performance on each single-objective preference-based subproblem.

**Theorem 1.** *Let $x_\lambda^*$ denotes the optimal solution of the problem (13) with preference $\lambda$, if the proposed method can generate an approximate solution $x_\lambda \prec_\varepsilon x_\lambda^*$ for any preference $\lambda$, it is able to generate an $\varepsilon$-approximate Pareto set $\mathcal{P}_\varepsilon$ to the MOCO problem.*

*Proof.* Let $\mathcal{P}$ be the Pareto set for a MOCO problem, for any $x_{\text{Pareto}} \in \mathcal{P}$, according to **Lemma 1**, there is a weight vector $\lambda > 0$ such that $x = x_\lambda^*$ is the optimal solution for subproblem (13) with a specific preference $\lambda$. Therefore, our proposed method can generate an approximated solution $x_\lambda \prec_\varepsilon x_\lambda^* = x_{\text{Pareto}}$. By generating approximate solutions for all $x_{\text{Pareto}} \in \mathcal{P}$, our proposed method is able to generate an $\varepsilon$-approximate Pareto set $\mathcal{P}_\varepsilon$ to the MOCO problem. □

## A.3 LIMITATION

**Strong Assumption on (Approximately) Solving all Subproblems:** The approximation guarantee in **Theorem 1** heavily depends on the ability to (approximately) solve each weighted subproblem. Due to the NP-harness, it is indeed non-trivial to give a convergence guarantee to generate $\varepsilon$-dominate solutions for any preference with a small enough $\varepsilon$. This limitation also applies for other end-to-end learning-based (e.g, neural combinatorial optimization) and heuristic-based methods.

We are aware that some efforts have been made to combine the learning-based method with dynamic programming to achieve asymptotically optimal solution solution for specific single-objective problem in recent works (Cappart et al., 2021b; Kool et al., 2021). These methods provide a controllable trade-off between the solution quality and the computational cost for solving NP-hard problems.

However, their generalization to the multi-objective problem is not straightforward, since the scalarized subproblem for each preference is not necessary the same as its single-objective counterpart. For example, a Tchebycheff scalarized MOTSP is not a single-objective TSP as discussed at the end of Section 3.2. In addition, according to Bengio et al. (2020), these methods belong to the class of learning alongside the algorithms, while our proposed approach is learning to directly produce the solutions (neural combinatorial optimization). Therefore, the idea for learning enhanced multi-objective combinatorial algorithm could be an important research topic in future, but out of the scope for the current work.

**Dense Approximation for the Whole Pareto Set:** Another concern would be the required number of solutions in the $\varepsilon$-approximate Pareto set $\mathcal{P}_\varepsilon$. If the required number is exponential to the input size, the approximation itself is also intractable. In their seminal work, Papadimitriou & Yannakakis (2000) establish a promising result:

**Theorem 2 (Papadimitriou & Yannakakis (2000)).** *For any multiobjective optimization problem and any $\varepsilon$, there is an $\varepsilon$-approximate Pareto set $\mathcal{P}_\varepsilon$ of which the size is polynomial in the number of solutions and $\frac{1}{\varepsilon}$ (but exponential in the number of objectives).*

However, the existence of such a set still does not mean that it can be easily found (Papadimitriou & Yannakakis, 2000; Herzel et al., 2021). The computability (whether $\mathcal{P}_\varepsilon$ can be constructed in polynomial time) would be hard to justify for a real-world problem. For a new unseen problem instance in practice, our proposed method might still need to generate an exponentially large number of solutions to construct an $\varepsilon$-approximate Pareto set $\mathcal{P}_\varepsilon$. It is also unclear how to properly select a set of preferences in advance. Many research efforts have been made on developing approximation methods for solving MOCO problems in the past decades (Herzel et al., 2021; Hansen, 1980; Papadimitriou & Yannakakis, 2000; Vassilvitskii & Yannakakis, 2005; Koltun & Papadimitriou, 2005; Bazgan et al., 2017). In future work, it is important to better leverage the current advanced approximation strategies to design more efficient preference-based methods. In the learning-based optimization scenario we consider, it is also possible to learn the suitable approximation method and/or preference distribution directly from the data (problem instances).

# B  DETAILS ON THE PROPOSED MODEL

## B.1  MODEL SETTING

Table 4: Model Information.

|  | #Models | #Total Params | #Encoder Params | #Decoder Params | #Preference |
|---|---|---|---|---|---|
| Attention Model | 1 | 1.3M | 1.2M | 0.1M | Fixed 1 |
| P-MOCO (Ours) | 1 | 1.4M | 1.2M | 0.2M | Flexible |

We use the same model for all MOCO problems while tuning the input size and mask method for each problem. Table 4 shows the number of parameters of a standard single-objective attention model (Kool et al., 2019) and our proposed preference-based multiobjective attention model. Our model supports flexible preference assignment at the inference time with a small overhead, while the other neural MOCO methods all require training multiple AM models for different preferences. We build the single-preference attention models as well as our model following the implementation in Kwon et al. (2020).

**Attention Encoder.** The encoder we use is the standard attention encoder as in Kool et al. (2019), and it is shared by all preferences. The encoder has 6 attention layers, and 128-dimensional node embedding for the input nodes. Each attention layer has a multi-head attention (MHA) with eight 16-dimensional heads, and a fully connected layer (FC) with one 512-dimension hidden sublayer. The encoder also includes skip-connection and batch normalization for each attention layer. We use the same model for all MOCO problems (MOTSP, MOCVRP, MOKP) but with different input dimensions for each problem, which will be introduced in the next section.

**Preference-Conditioned Decoder.** The decoder's main model structure is the same as the AM decoder (Kool et al., 2019). It has one multi-head attention layer with eight 16-dimensional heads similar to the encoder, but without skip-connection and batch normalization. The decoder uses a single 128-dimensional attention head to calculate the probabilities of selecting different nodes at each step. Different problems have different masking methods for probability calculation.

We use a simple MLP model to generate the preference-conditioned parameters for the decoder. For all MOCO problems, the MLP model has two 128-dimensional hidden layers with ReLu activation. The input is an $m$-dimensional preference vector $\lambda$ which satisfies $\lambda_i \geq 0$ and $\sum_{i=1}^m \lambda_i = 1$, where $m$ is the number of objectives and $\lambda_i$ is the preference for the $i$-th objective. We adopt the parameter compression approach in Ha et al. (2017) to control the model size. The MLP model first generates a hidden embedding $e(\lambda) = \mathbf{MLP}(\lambda|\psi)$, then maps the hidden embedding to the decoder parameters via linear projection $\theta_{\mathbf{decoder}} = We(\lambda) + b$. The learnable parameters are $\psi$ for the MLP model $\mathbf{MLP}(\lambda|\psi)$ and the parameter matrices $W$ and $b$ for the decoder.

**Training Procedure.** For all problems, we train our proposed model for 200 epochs, with $100,000$ problem instances randomly generated on the fly at each epoch. At each iteration step, we need to sample $K$ preferences, $B$ problem instances, and $N$ tours to calculate the policy gradient. We set $K \times B = 64$ to make the batch of 64 instances for training a single AM model, and let $N$ equal to the problem size (e.g., the number of nodes) as in Kwon et al. (2020). We find the model performance is equally good for setting $K = 1, 2$ and $4$, and keep using $K = 1$ for all problems. In other words, we randomly generate a preference $\lambda$ that satisfies $\lambda_i \geq 0$ and $\sum_{i=1}^m \lambda_i = 1$ at each training step.

For the AM-MOCO baseline, we adapt the transfer training approach in Li et al. (2020) to train multiple AM models for different preferences. We first train a single AM model with a single preference on one objective from scratch with 200 epochs, then transfer its parameter to the model for neighbor subproblem with similar preference, and fine-tune the new model with 5 epochs. With sequentially transfer and fine-tune, we can obtain a set of trained models for different preferences. In most experiments, we set the number of preferences as 101. Therefore, we need to build 101 AM models with total 700 training epochs.

**Instance Augmentation for MOCO.** Due to the design choice of minimal essential change (e.g., the preference-conditioned decoder), our method can also enjoy the current improvements that were originally proposed for the single objective NCO. Here, we generalize the instance augmentation method proposed in Kwon et al. (2020) to the MOCO version.

The key idea of instance augmentation for NCO is to find multiple efficient transformations for the original problem such that they share the same optimal solution. Then, we can use an NCO method to solve all problems and select the best solution among all obtained (potentially different) solutions. In this way, we have a more robust result similar to the test-time augmentation for computer vision (Szegedy et al., 2016). For the single-objective euclidean TSP and CVRP, there is a set of straightforward transformations, which simply flips or rotates the coordinate for all the 2D locations in a problem instance (Kwon et al., 2020). For a location $(x, y)$, there is eight different transformation, namely, $\{(x,y), (y,x), (x,1-y), (y,1-x), (1-x,y), (1-y,x), (1-x,1-y), (1-y,1-x)\}$.

For an $m$-objective euclidean MOTSP problem, the concrete location representations are independent for each objective. Therefore, we can independently apply different transformations for each objective. Consider the above eight different transformations for each objective, we can have $8^m$ different problem transformations for an MOTSP instance. We have fixed $8$ transformations for MOCVRP since it only has one 2D coordinate, and no transformation for MOKP. The details for each problem can be found in the next section.

## B.2  Training Efficiency

We use the same amount of samples to train our proposed preference-based model as the other single-objective solvers need (Kool et al., 2019; Kwon et al., 2020). Indeed, our proposed model requires significantly fewer samples and training epochs, compared to the other MOCO methods that need to build multiple models for different preferences.

Table 5: The Performance on a Single Preference.

|       | Concorde | LKH | OR Tools | AM (Single Obj.) | P-MOCO (Single Pref.) | P-MOCO (all pref.) |
|-------|----------|-----|----------|------------------|-----------------------|--------------------|
| TSP20  | 3.83 (5m)  | 3.83 (42s) | 3.86 (1m)  | 3.83 (4s)  | 3.83 (4s)  | 3.83 (4s)  |
| TSP50  | 5.69 (13m) | 5.69 (6m)  | 5.85 (5m)  | 5.71 (15s) | 5.71 (15s) | 5.71 (15s) |
| TSP100 | 7.76 (1h)  | 7.76 (25m) | 8.06 (23m) | 7.82 (1m)  | 7.82 (1m)  | 7.82 (1m)  |

We compare our model's performance on one of the objective (e.g., with preference $(1,0)$) with the other SOTA single-objective solver and learning-based solver, the results are shown in Table 5. The results of Concorde/LKH/OR Tools are from Kwon et al. (2020), and we run the learning-based solver by ourselves. We report the average performance over $10,000$ test instances. AM is the single-objective solver (one model in AM-MOCO), P-MOCO (single preference) is our proposed model but only training on a single fixed preference $(1,0)$, and P-MOCO (all preferences) is our proposed model with the reported result on the preference $(1,0)$. With the same amount of training samples, our model has similar single-objective performance with learning-based single-objective solver, while it can additionally approximate the whole Pareto front. The learning-based solver's performance can be further improved by sampling or active search.

These results indicate that we can use a single encoder to efficiently learn a shared representation for all trade-offs among different objectives, and there is a positive knowledge transfer among preferences during the learning procedure. In addition, it also confirms the assumption that similar preferences should have similar corresponding (approximate) Pareto solutions for the multiobjective problems we consider in this paper. These findings could be useful to design more powerful learning-based models for MOCO in the future.

## B.3  Active Adaption

After end-to-end training, our proposed method can directly generate different trade-off solutions to a given problem without further search procedure. However, similar to single-objective neural combinatorial optimization, this approach could still have a gap to the Pareto front, especially for problems out of the training distribution $\mathcal{S}$ (e.g., with different sizes and patterns) (Lisicki et al., 2020). Iterative search methods, such as sampling and beam search, can further improve the performance for a single solution or single preference (Veličković & Blundell, 2021). However, these approaches can not find a better approximation to the whole Pareto set for a MOCO problem.

---

**Algorithm 2** Neural MOCO Active Adaption

---

1: **Input:** model parameter $\boldsymbol{\theta}$, instance $s$, preference distribution $\Lambda$, number of adaption steps $T$, number of preferences per iteration $K$, number of tours $N$
2: **for** $t = 1$ to $T$ **do**
3:     $\lambda_k \sim$ **SamplePreference**$(\Lambda)$    $\forall k \in \{1, \cdots, K\}$
4:     $\boldsymbol{\pi}_k^j \sim$ **SampleTour**$(p_{\boldsymbol{\theta}(\lambda_k)}(\cdot|s))$    $\forall k \in \{1, \cdots, K\}$    $\forall j \in \{1, \cdots, N\}$
5:     $b(s|\lambda_k) \leftarrow \frac{1}{N} \sum_{j=1}^{N} L(\boldsymbol{\pi}_k^j|\lambda_k, s)$    $\forall k \in \{1, \cdots, K\}$
6:     $\nabla \mathcal{J}(\boldsymbol{\theta}) \leftarrow \frac{1}{KN} \sum_{k=1}^{K} \sum_{j=1}^{N} [(L(\boldsymbol{\pi}_k^j|\lambda_k, s) - b(s|\lambda_k))\nabla_{\boldsymbol{\theta}(\lambda_k)} \log p_{\boldsymbol{\theta}(\lambda_k)}(\boldsymbol{\pi}_k^j|s)]$
7:     $\theta \leftarrow$ **ADAM**$(\theta, \nabla \mathcal{J}(\boldsymbol{\theta}))$
8: **end for**
9: **Output:** The model parameter $\boldsymbol{\theta}$

---

We propose a simple yet powerful active adaption approach as shown in **Algorithm 2**. It iteratively adapts the model parameter $\boldsymbol{\theta}(\lambda)$ to a given instance $s$ (or a batch of instances) with all preferences from the distribution $\Lambda$ rather than searching for a specific solution. This method is similar to the active search in Bello et al. (2017) which actively refines the single-objective model for efficient candidate solutions searching. Our approach focuses on adapting the whole model for a better Pareto front approximation. Since this method is distribution-agnostic (not depend on specific instance distribution $\mathcal{S}$), it is suitable for out-of-distribution adaption.

## C  DETAILS OF THE MOCO PROBLEMS

This section introduces the detailed problem formulation for the MOTSP, MOCVRP and MOKP we used in this work. We also provide the model configuration (e.g., input size, masks) for each problem.

### C.1  MOTSP

We consider the Euclidean multiobjective traveling salesman problem (Euclidean MOTSP), which is widely used in the MOCO community (Lust & Teghem, 2010b; Florios & Mavrotas, 2014). Its single objective counterpart, 2D Euclidean TSP, has also been studied in single-objective neural combinatorial optimization (NCO) (Vinyals et al., 2015; Bello et al., 2017; Kool et al., 2019). A general $m$-objective MOTSP instance $s$ with $n$ nodes has $m$ $n \times n$ cost matrices $\{C^i = (c^i_{jk}), i = 1, \cdots, m\}$ for $m$ different costs. The problem is to find a tour (cyclic permutation $\boldsymbol{\pi}$) to minimize all the costs:

$$\min L(\boldsymbol{\pi}|s) = \min(L_1(\boldsymbol{\pi}|s), L_2(\boldsymbol{\pi}|s), \cdots, L_m(\boldsymbol{\pi}|s)),$$

$$\text{where } L_i(\boldsymbol{\pi}|s) = c^i_{\boldsymbol{\pi}(n)\boldsymbol{\pi}(1)} + \sum_{j=1}^{n-1} c^i_{\boldsymbol{\pi}(j)\boldsymbol{\pi}(j+1)}. \tag{14}$$

In a Euclidean MOTSP, the cost information is stored in the nodes rather than the edges. The $j$-th node has a $2m$-dimensional vector $[\boldsymbol{x}^1_j, \boldsymbol{x}^2_j, \cdots, \boldsymbol{x}^m_j]$ where $\boldsymbol{x}^i_j \in \mathbb{R}^2$ is a 2D coordinate for the $i$-th objective. The $i$-th cost $c^i_{jk} = ||\boldsymbol{x}^i_j - \boldsymbol{x}^i_k||_2$ is the Euclidean distance for moving from node $j$ to $k$.

If we only have one objective $m = 1$, it reduces to the single-objective 2D Euclidean TSP:

$$\min_{\boldsymbol{\pi}} L_1(\boldsymbol{\pi}|s) = ||\boldsymbol{x}_{\boldsymbol{\pi}(n)} - \boldsymbol{x}_{\boldsymbol{\pi}(1)}||_2 + \sum_{j=1}^{n-1} ||\boldsymbol{x}_{\boldsymbol{\pi}(i)} - \boldsymbol{x}_{\boldsymbol{\pi}(i+1)}||_2. \tag{15}$$

The single-objective TSP is already NP-hard, so does the MOTSP. In addition, the Pareto set of MOTSP has an exponential cardinality with respect to its input size (e.g., number of nodes), so it is intractable even for the 2-objective case (Ehrgott & Gandibleux, 2003).

**Problem Instance.** Similar to the previous work on single-objective NCO (Lust & Teghem, 2010b; Florios & Mavrotas, 2014), we randomly sample all $n$ nodes with uniform distribution on the $2m$-dimensional unit hyper-square (e.g., $[0, 1]^{2m}$) for all problem instances.

**Model Details.** In $m$-objective MOTSP, each node has a $2m$-dimensional vector to store all cost information, so the input size is $2m$ for the encoder. To calculate the probability for selecting the next node, the decoder needs to mask all already visited nodes as unavailable. We have a valid tour when all node is selected (we assume the end node will connect to the start node).

### C.2  MOCVRP

The vehicle routing problem (VRP) is a classical generalization of TSP, which has been studied for several decades. This work studies the capacitated vehicle routing problem (CVRP). In this problem, in addition to the location, each node (city) has a demand $\delta_i$ needed to be satisfied. There is an extra depot node and a vehicle with a fixed capacity $D > \delta_i, \forall i$ to handle all the demands. The vehicle will always start from the depot node, then goes to different cities to satisfy multiple demands $\sum \delta_i \leq D$, and turns back to the depot node. A solution to this problem is a set of routes that satisfies the demands for all cities.

In the multiobjective problem, we consider two objectives to optimize. The first one is the total tour length as in the single-objective CVRP, and the other one is the tour length for the longest route (which is also called makespan in scheduling theory). This problem has been studied in the MOCO community (Lacomme et al., 2006).

**Problem Instance.** Similar to the TSP problem, the location of $n$ nodes are uniformly sampled from the unit square. For the demand, similar to the previous work on the single-objective counterpart (Kool et al., 2019; Kwon et al., 2020), we randomly sample discrete $\delta_i$ from the set $\{1, \cdots, 9\}$. For problem with size $n = 20, 50, 100$, we set the capacity as $D_{20} = 30, D_{50} = 40$

and $D_{100} = 50$, respectively. Without loss of generality, we normalize the demands $\hat{\delta}_i = \frac{\delta_i}{D}$ and capacity $\hat{D} = \frac{D}{D} = 1$ as in the previous work (Kool et al., 2019; Kwon et al., 2020). Split delivery is not allowed in this problem.

**Model Details.** In the MOCVRP, the depot node has a 2-dimensional location vector, and the other nodes all have 3-dimensional vectors to store their locations and demands. We use different parameter matrices to project the nodes into the input embedding with the same dimension $d_h = 128$. For node selection, the model records the current capacity of the vehicle and the rest demands for all nodes. If a node has been already visited or has demand larger than the vehicle's current capacity, it will be masked as unavailable for the vehicle to visit. If no node is available to visit, the vehicle will go back to the depot. Once all nodes have 0 demands, the node selection is finished and we have a valid solution to the problem.

## C.3 MOKP

Knapsack problem (KP) is also a widely studied combinatorial optimization problem. In this work, we consider the 0-1 multiobjective knapsack problem (MOKP) with $m$ objectives and $n$ items:

$$
\begin{aligned}
&\max f(x) = \max(f_1(x), f_2(x), \cdots, f_m(x)), \\
&\text{where } f_i(x) = \sum_{j=1}^n v_j^i x_j, \\
&\text{subject to } \sum_{j=1}^n w_j x_j \le W, \quad x_j \in \{0, 1\},
\end{aligned}
\tag{16}
$$

where each item has a weight $w_j$ and $m$ different values $\{v_j^i, i = 1, \cdots, m\}$. The problem (e.g., knapsack) has a maximum weight capacity $W$, and the goal is to select a set of items within the weight capacity to maximize the sum values for each objective. To make this problem nontrivial, we further assume all values $v_j^i, \forall i, j$, weights $w_j \forall j$ and the total capacity are non-negative real value. The total weight of all items is larger than the capacity $\sum w_i > W$, while each single weight is smaller than the capacity $w_i < W, \forall i = 1, \cdots, n$. The single-objective knapsack problem is NP-hard, so does the MOKP problem (Ehrgott & Gandibleux, 2003).

**Problem Instance.** We randomly generate the values and weight for each item both uniformly in $[0, 1]$. We consider problems with $n = 50, 100, 200$ nodes, and the weight capacities are $W_{50} = 12.5, W_{100} = W_{200} = 25$ as in the previous work (Bello et al., 2017; Kwon et al., 2020).

**Model Details.** In an $m$-objective MOKP, each item has $m$ values and 1 weight, so the input dimension is 3 for the encoder. For node selection at each step, we mask all already selected nodes and nodes with weights larger than the remained capacity as unavailable. We terminate the selection when all nodes are labeled as unavailable.

# D ADDITIONAL EXPERIMENTAL RESULTS

## D.1 HYPERVOLUME INDICATOR

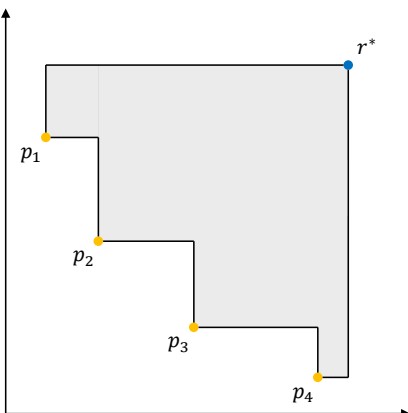

Figure 4: Hypervolume illustration.

To solve a MOCO problem, the result for each method is a set of approximate Pareto solutions. Since the ground truth Pareto set is usually unknown, we use the hypervolume (HV) indicator (Zitzler et al., 2007) to numerically compare the performance for each method. The hypervolume indicator is widely used in the MOCO community for algorithm comparison.

The hypervolume of a set is the volume in the objective space it dominates. For a set $P \subset \mathbb{R}^m$ in the objective space, we can find a reference point $r^*$ that dominated by all solutions in $P$, and define the hypervolume $\text{HV}(P)$ as the volume of the set:

$$S = \{r \in \mathbb{R}^m \mid \exists y \in P \text{ such that } y \prec r \prec r^*\}, \tag{17}$$

where $\text{HV}(P) = \textbf{Vol}(S)$. An illustration example is shown in Figure 4. The grey area is the set $S$ dominated by the solutions in set $P = \{p_1, p_2, p_3, p_4\}$ with the reference point $r^*$. In this 2-dimensional case, the hypervolume $\text{HV}(P)$ is the size of the grey area.

The hypervolume indicator has two important advantages for measuring the approximate set quality with respect to Pareto optimality (Zitzler et al., 2007). First, if an approximate set $A$ dominates another approximate set $B$, it will have a strictly better hypervolume $\text{HV}(A) > \text{HV}(B)$. In addition, if an approximate set $C$ contains all Pareto optimal solutions, it is guaranteed to have the maximum hypervolume value. In comparison, an approximate set has better performance if it has a larger hypervolume.

With different objective scales, the hypervolume value will vary significantly among different problems. We report the normalized hypervolume values $\hat{H}(P) = \text{HV}(P)/\prod_i^m r_i^*$ for all methods and also their performance gaps to our method. For each experiment, all methods share the same reference point $r^*$, which contains the largest value achieved for each objective. Since all problems we consider have positive objective values, we have $0 \leq \hat{H}(P) \leq 1$ for all solution sets. The ground truth Pareto set $P^*$ usually has $\hat{H}(P^*) < 1$, unless the zero vector $\mathbf{0} \in \mathbb{R}^m$ is feasible and in the Pareto set.

## D.2 OUT-OF-DISTRIBUTION PROBLEM WITH EXACT PARETO FRONT

Table 6: The Results on Problems with Exact Pareto Front.

| | L1 | | | L2 | | |
|---|---|---|---|---|---|---|
| Method | HV | HV Gap | IGD | HV | HV Gap | IGD |
| Exact Pareto front | 0.733 | - | 0 | 0.735 | - | 0 |
| Weight-Sum LKH | 0.728 | 0.71% | 0.008 | 0.730 | 0.75% | 0.007 |
| Weight-Sum OR Tools | 0.721 | 1.68% | 0.012 | 0.722 | 1.76% | 0.009 |
| P-MOCO (101 pref.) | 0.718 | 2.01% | 0.014 | 0.717 | 2.43% | 0.013 |
| P-MOCO (101 pref., aug.) | 0.722 | 1.52% | 0.012 | 0.721 | 1.88% | 0.010 |
| P-MOCO (101 pref., aug., active) | 0.724 | 1.21% | 0.010 | 0.724 | 1.48% | 0.008 |
| | L3 | | | L4 | | |
| Method | HV | HV Gap | IGD | HV | HV Gap | IGD |
| Exact Pareto front | 0.737 | - | 0 | 0.737 | - | 0 |
| Weight-Sum LKH | 0.733 | 0.53% | 0.007 | 0.731 | 0.83% | 0.006 |
| Weight-Sum OR Tools | 0.723 | 1.89% | 0.012 | 0.718 | 2.56% | 0.014 |
| P-MOCO (101 pref.) | 0.722 | 2.04% | 0.013 | 0.720 | 2.27% | 0.012 |
| P-MOCO (101 pref., aug.) | 0.724 | 1.72% | 0.012 | 0.723 | 1.93% | 0.011 |
| P-MOCO (101 pref., aug., active) | 0.726 | 1.43% | 0.010 | 0.727 | 1.33% | 0.008 |
| | L5 | | | L6 | | |
| Method | HV | HV Gap | IGD | HV | HV Gap | IGD |
| Exact Pareto front | 0.734 | - | 0 | 0.746 | - | 0 |
| Weight-Sum LKH | 0.727 | 0.92% | 0.007 | 0.742 | 0.48% | 0.006 |
| Weight-Sum OR Tools | 0.721 | 1.79% | 0.012 | 0.728 | 2.43% | 0.015 |
| P-MOCO (101 pref.) | 0.719 | 2.05% | 0.014 | 0.729 | 2.19% | 0.014 |
| P-MOCO (101 pref., aug.) | 0.723 | 1.48% | 0.011 | 0.732 | 1.82% | 0.012 |
| P-MOCO (101 pref., aug., active) | 0.725 | 1.29% | 0.009 | 0.737 | 1.19% | 0.009 |

We conduct experiments on 6 two-objective MOTSP100 instance (L1-L6) in Florios & Mavrotas (2014) of which the exact Pareto fronts are available. In these problems, the objective functions have different ranges, and the cities are not uniformly located, so they are out of our method's training distribution. The results can be found in the Table 6.

In addition to hypervolume, we also report the Inverted Generational Distance (IGD) (Fonseca et al., 2006) to measure the average Euclidean distance between the set of approximated Pareto solutions to the exact Pareto front. A smaller IGD value means the approximated set is closer to the exact Pareto front. According to the results, our method, with the instance augmentation and/or active search (10 min budget), can have a good performance on these out-of-distribution (OOD) instances with a $1\% - 1.5\%$ hypervolume gap. The proposed method also significantly outperforms the weight-sum OR tools baseline. There is still a gap to the strong weight-sum LKH baseline. As discussed in the paper, robust OOD generalization is an important research direction for the learning-based solver.

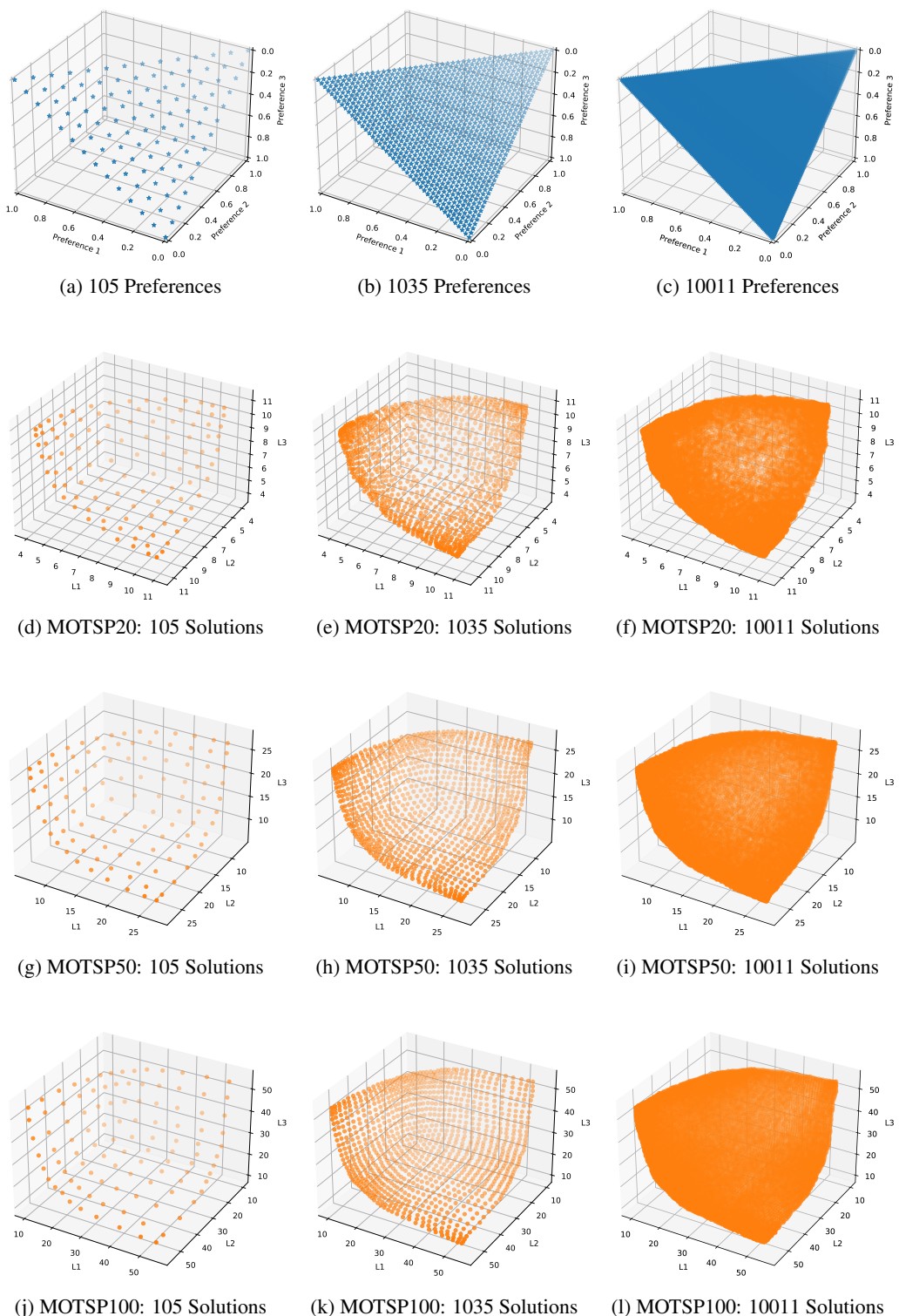

Figure 5: Different number of uniform distributed preferences and the corresponding solutions generated by our method on MOTSP20, MOTSP50 and MOTSP100. Our model can generate well-distributed solutions with a small number of preferences, and generate a dense approximation with a large number of preferences.

### D.3 FLEXIBLE PREFERENCE-BASED APPROXIMATION

With our model, it is flexible to generate different number of solutions to approximate the Pareto front. We present an example on the three-objective TSP in Figure 5. We use the structured weight assignment approach from Das & Dennis (1998) to give the sets of weights for different instances. This method can generate $n = C_p^{m+p-1}$ evenly distributed weights with an identical distance to their nearest neighbor on the unit simplex (e.g., $\sum_{i=1}^{m} \lambda_i = 1$ with $\lambda_i \geq 0, \forall i$), where $m$ is the number of objectives and $p$ is a parameter to control the number of weights.

For the three objective TSP problems ($m = 3$), we assign $p = 13, 44$ and $140$ to generate $n = 105, 1035$ and $10011$ weights respectively. We also show the corresponding generated solutions for MOTSP instances with $20, 50$ and $100$ cities. According to the results in Figure 5, our model can generate well-distributed solutions with a small number of preferences, and generate a dense approximation with more preferences. The ability to generate a dense approximation to the whole Pareto set also allows the decision-maker to generate arbitrary preferred solutions on the approximate front.

### D.4 PREFERENCE-SOLUTION CONNECTION

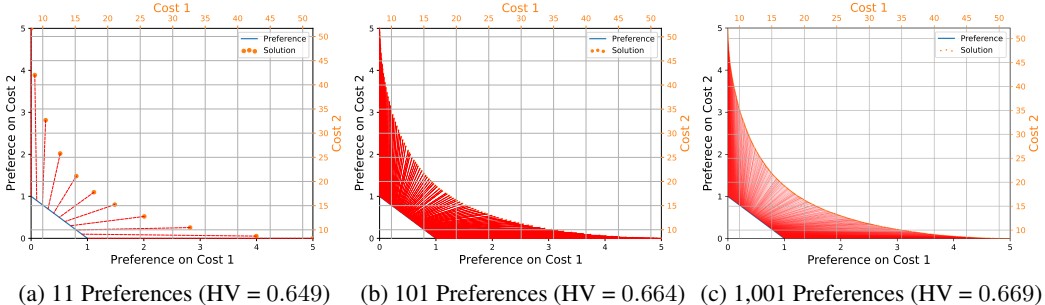

(a) 11 Preferences (HV = 0.649)     (b) 101 Preferences (HV = 0.664)     (c) 1,001 Preferences (HV = 0.669)

Figure 6: Different number of uniformly distributed preferences and their connections to the corresponding solutions on MOTSP100.

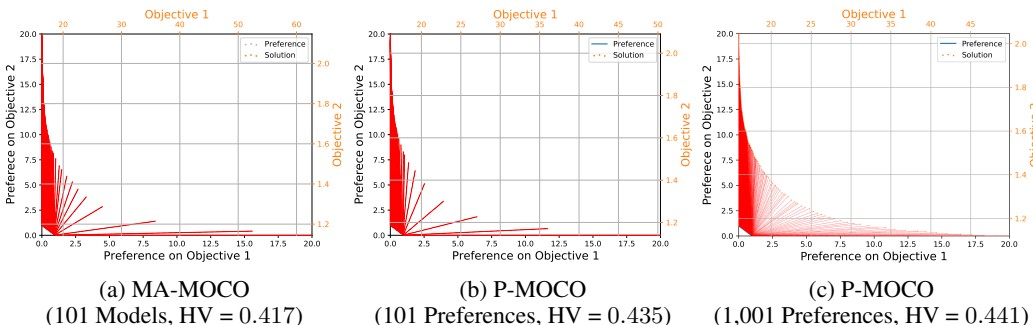

(a) MA-MOCO
(101 Models, HV = 0.417)

(b) P-MOCO
(101 Preferences, HV = 0.435)

(c) P-MOCO
(1,001 Preferences, HV = 0.441)

Figure 7: Different number of uniformly distributed preferences and their connections to the corresponding solutions on MOCVRP100. We report the approximate Pareto front for MA-MOCO with 101 models, and our model with 101 and 1, 001 preferences.

We further analyze the connection between the preference and its corresponding solution on the uniform and non-uniform Pareto front. Figure 6 shows the connections in our model with different numbers of preferences for the MOTSP100 instance. Since the two objectives (costs) in MOTSP have the same scale, this problem has a uniform connection between the preferences and the (approximate) Pareto front. By increasing the number of preferences, we have three sparse to dense generated Pareto front approximations.

We are more interested in MOCVRP, which has a non-uniform Pareto front. In this problem, we consider two different objectives to optimize, namely, the total tour length (objective 1) and the tour

length for the longest route (objective 2). These two objectives are in quite different scales, where the first objective is significantly larger than the second one. In Figure 7, we show different connections for the MOCVRP100 instance. For MA-MOCO, we report the connections for all 101 models. For our proposed model, we report the connections with different numbers of uniform preferences.

In this problem, 101 models or our model with 101 uniform preferences are not enough to generate a dense approximate Pareto front. The obtained solutions are biased to the area that objective 1 has a much better relative performance. By increasing the number of preferences, our proposed method can generate more solutions that have relatively better performance for objective 2, which leads to a better Pareto front approximation with higher hypervolume. In this work, we always use a straightforward uniform sampling method to select the preferences. It is interesting to design a learning-based approach to select the preferences for a given problem instance. Preference adjustment and model adaption with awareness on the shape of Pareto front are also worthy to investigate. We left them to the future work.

In the MOCVRP instance, we also find the 101-model MA-MOCO has a worse performance compared to our method with 101 preferences. The reason would be the mismatch between the uniform transfer training and the non-uniform Pareto front. Increasing the training steps for fine-tuning each model might fix this issue, but will lead to an even larger computational overhead, given the current training already require 700 epochs. The fixed preferences assignment is another issue for MA-MOCO. It requires a fixed set of preferences for each model at the start of the training procedure when the decision makers might have no knowledge on the problem. When the training procedure is done, it dose not allow any preference adjustment without retraining the models.

### D.5 Connection between Preferences and Solutions

In the previous sections, we use the weighted Tchebycheff aggregation to connect the preference to its corresponding solution for two-objective optimization problems:

$$g_{\text{tch}}(x|\lambda) = \max_{1 \leq i \leq m} \{\lambda_i |f_i(x) - z_i^*|\}, \tag{18}$$

where $z_i^* < \min_{x \in \mathcal{X}} f_i(x)$ is an ideal value for $f_i(x)$. There are also many other aggregation function we can use to build the connection. For example, a modified version of weighted Tchebycheff aggregation can be defiend as:

$$g_{\text{mtch}}(x|\lambda) = \max_{1 \leq i \leq m} \{\frac{1}{\lambda_i} |f_i(x) - z_i^*|\}, \tag{19}$$

where the only difference is the weight vector $\frac{1}{\lambda_i}$.

The penalty-based boundary intersection (PBI) is another widely-used aggregation function for decomposition-based multiobjective optimization (Zhang & Li, 2007):

$$
\begin{aligned}
g_{\text{pbi}}(x|\lambda) &= d_1 + \theta d_2, \\
d_1 &= |(F(x) - z^*)^T \lambda| / ||\lambda||, \\
d_2 &= ||F(x) - z^* - d_1 \frac{\lambda}{||\lambda||}||,
\end{aligned}
\tag{20}
$$

where $\theta$ is the penalty parameter, $F(x) = (f_1(x), \ldots, f_m(x))$ and $z^* = (z_i^*, \ldots, z_i^*)$ are the objective vector and ideal vector respectively. An inverted version of PBI (IPBI) aggregation function (Sato, 2014) can be defined as:

$$
\begin{aligned}
g_{\text{ipbi}}(x|\lambda) &= -d_1 + \theta d_2, \\
d_1 &= |(z^N - F(x))^T \lambda| / ||\lambda||, \\
d_2 &= ||z^N - F(x) - d_1 \frac{\lambda}{||\lambda||}||,
\end{aligned}
\tag{21}
$$

where $z^N$ is the nadir vector that contain each objective's worst value among all Pareto solutions.

For a two-objective optimization problem, when we can find a dense set of corresponding solutions to cover the Pareto front for each aggregation function, their performance could be similar to each

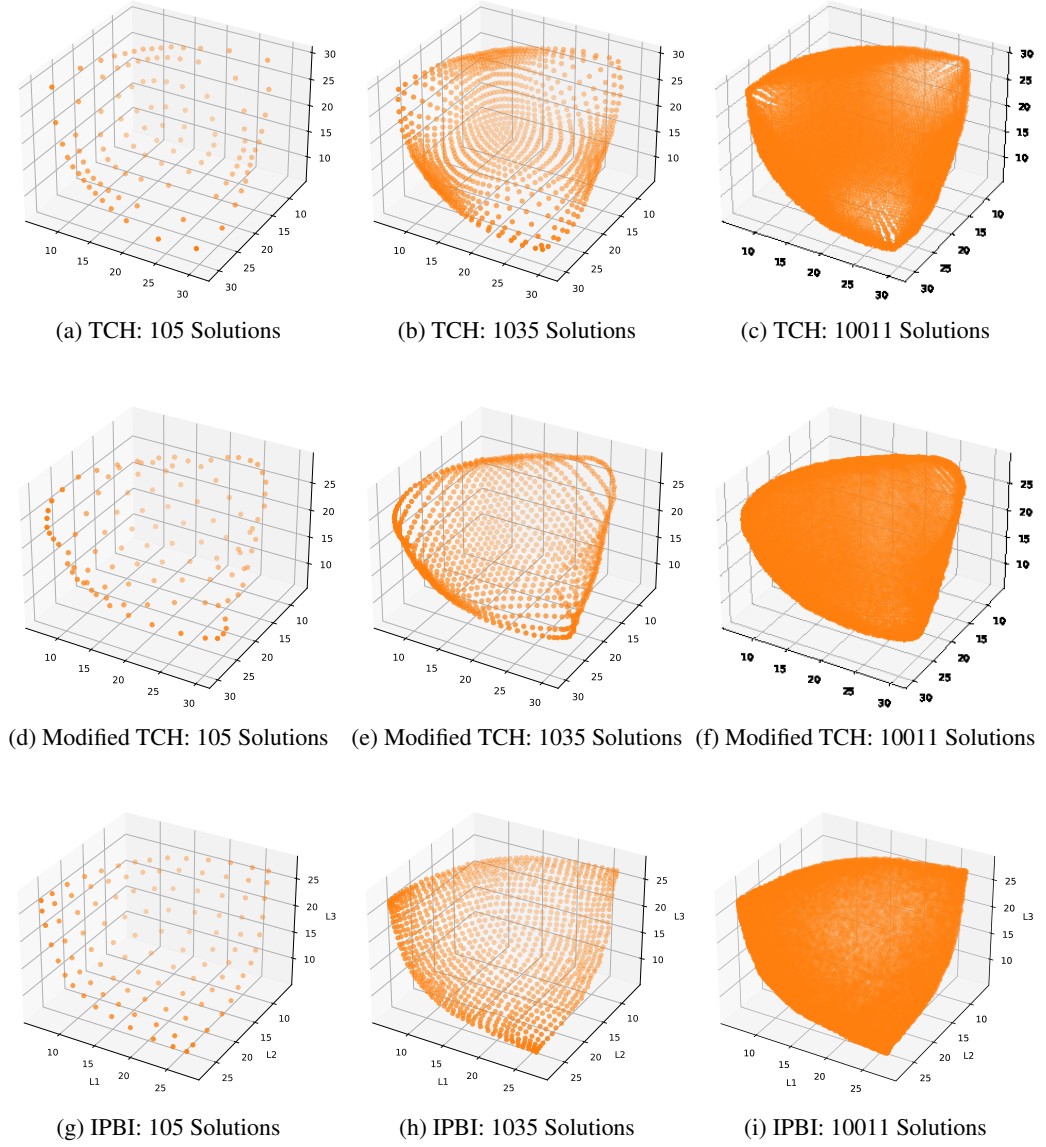

Figure 8: Different number of approximate Pareto solutions generated by our method with uniformly distributed weight vectors and different aggregation functions.

other. However, different aggregation functions would have quite different performances on the problems with three or more objective functions (called many-objective optimization problems). The performances will heavily depend on the shape of Pareto front (Ishibuchi et al., 2016), especially with a limited number of approximate solutions.

We compare the performance of our proposed method with different aggregation functions on MOTSP50 with 105, 1035 and 10011 preferences respectively in Fig. 8. According to the results, the IPBI method can generate the most uniformly distributed solutions for the MOTSP problem with an inverted triangular shape of Pareto front, of which the shape is similar to the weight vector distribution (e.g., see Fig 5). This observation is consistent with the findings and analysis in Ishibuchi et al. (2016). According to these results, we use the Tchebycheff aggregation for all two-objective optimization problems and IPBI aggregation for all problems with more than two objective functions in this work. Since the shape of Pareto front tends to be irregular for real-world applications (Ishibuchi et al., 2019), how to properly choose the aggregation function and assign the preference distribution could be an important future work.

## D.6 Three-Objective MOTSP with Asymmetric Pareto Front

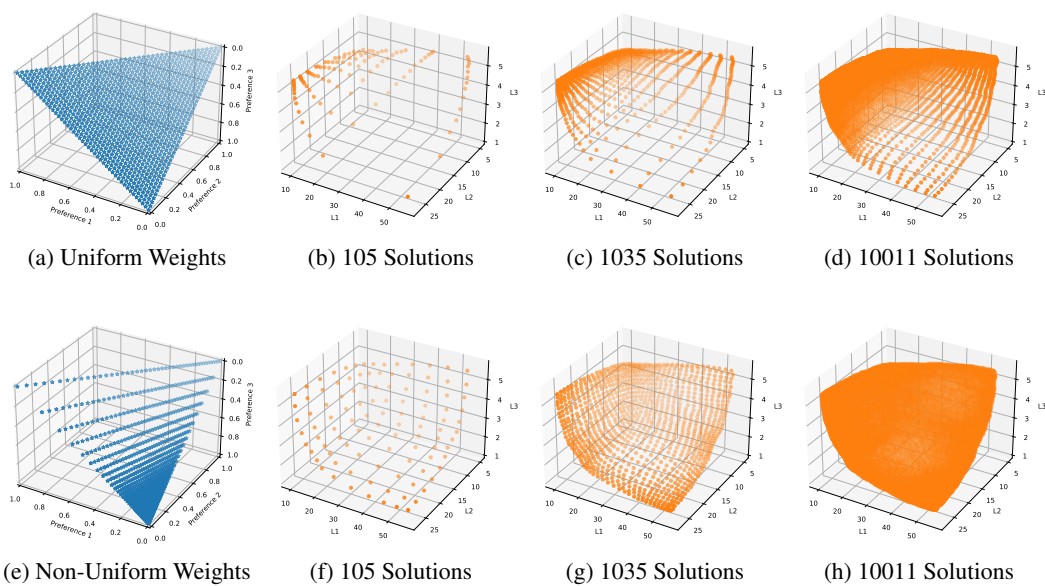

Figure 9: The uniform/non-uniform distributed preferences and different number of corresponding solutions generated by our method on three-objective MOTSP100 with asymmetric Pareto front. We use 1035 different preferences as example for the uniform and non-uniform case respectively, and present the results for 105, 1035 and 10011 solutions. **Top Row:** Uniform distributed preferences and the corresponding non evenly distributed solutions. **Bottom Row:** Non-uniform distributed preferences and the corresponding solutions which are more evenly distributed.

In this subsection, we conduct experiments on the three-objective MOTSP100 instances with asymmetric Pareto fronts. The definition of irregular MOTSP instance is almost the same as in Section C.1, except the coordinates for the three objectives and randomly sampled from $[0, 1]^2$, $[0, 0.5]^2$ and $[0, 0.1]^2$ respectively, rather than uniformly from $[0, 1]^6$. In this way, the objective values for the MOTSP instance will be in quite different scales, thus leading to an irregular Pareto front (the axes in Figure 9 are in different scales).

A well-known drawback of the scalarization-based approach is that it cannot evenly explore the irregular Pareto front with a set of uniform weights, which can also be observed in Figure 9(a)-(d). Our proposed approach allows the user to generate arbitrary trade-off Pareto solutions on the inference time, therefore they can directly generate a dense approximation and then select the preferred solutions as in Figure 9(d). This flexibility can partially address the unevenly distributed issues caused by a (small) set of fixed weights in the traditional scalarization-based approach.

If we know the approximate range of different objectives in advance, we can first normalize them into $[0, 1]$ to encourage a more symmetric Pareto front. Otherwise, on the inference time, we can use a (prior knowledge-based) biased and non-uniform weight assignment to generate uniformly distributed solutions. In Figure 9(e)-(h), we first multiple the three-dimensional weights by $(1, 2, 10)$ and then normalize them back to $[0, 1]^3$ which leads to a set of non-uniform weights as shown in Figure 9(e). With this weight assignment, we have a a set of more evenly distributed Pareto solutions as shown in Figure 9(f)-(h).

## D.7 Preference-based Inference

Even without any prior knowledge, our proposed approach allows the user to adaptively adjust the weights in real-time to search for the most suitable solutions in their preferred region(s). Some examples of selected weights and their corresponding solutions are shown in Figure 10 for symmetric Pareto front and Figure 11 for asymmetric Pareto front. If we have prior knowledge of the preference

(e.g., the decision-makers will only care about a specific region of the Pareto front), we can modify the training preference distribution Λ accordingly to enhance the training efficiency.

For the problem with a truly irregular Pareto front, it is also possible to adaptively adjust the given weights to make them evenly explore the Pareto front during the learning/searching process. One potential direction could be to consider the connection between scalarization and hypervolume maximization as in Zhang & Golovin (2020). We believe this could be an important research topic for the learning-based scalarization approach in future work.

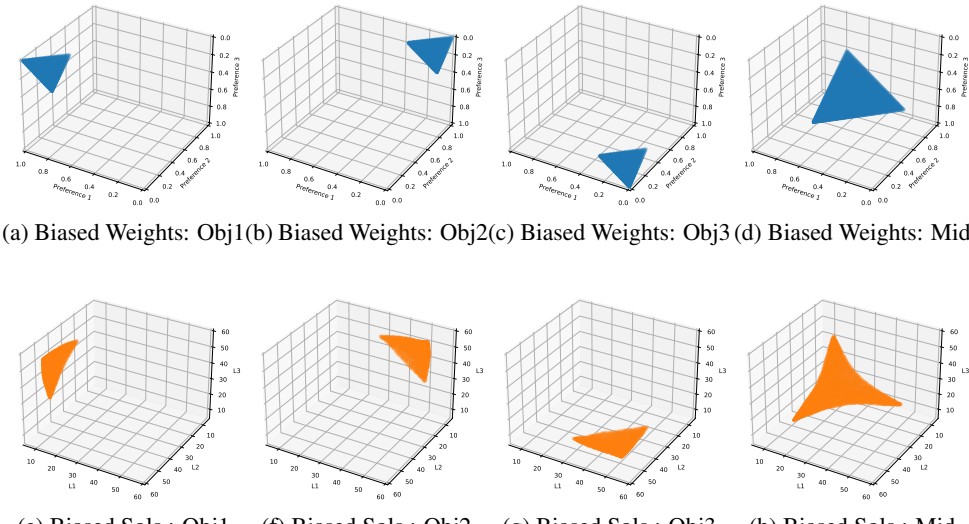

(a) Biased Weights: Obj1 (b) Biased Weights: Obj2 (c) Biased Weights: Obj3 (d) Biased Weights: Mid

(e) Biased Sols.: Obj1 (f) Biased Sols.: Obj2 (g) Biased Sols.: Obj3 (h) Biased Sols.: Mid

Figure 10: **Symmetric Pareto Front.** Different set of biased weights and their corresponding solutions on different regions of the symmetric Pareto front.

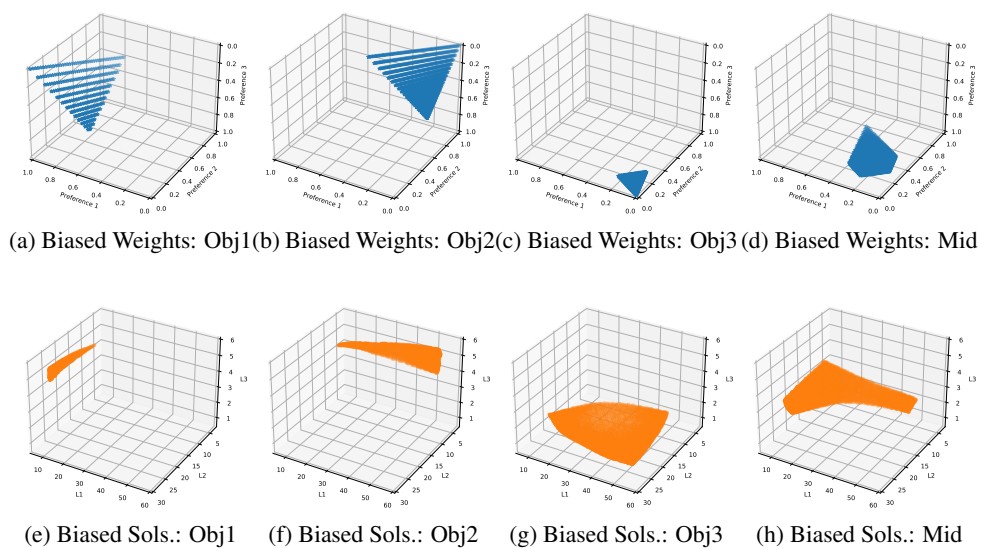

(a) Biased Weights: Obj1 (b) Biased Weights: Obj2 (c) Biased Weights: Obj3 (d) Biased Weights: Mid

(e) Biased Sols.: Obj1 (f) Biased Sols.: Obj2 (g) Biased Sols.: Obj3 (h) Biased Sols.: Mid

Figure 11: **Asymmetric Pareto Front.** Different set of biased weights and their corresponding solutions on different regions of the irregular Pareto front. The uniformness of the weights and the corresponding solutions could be different due to the asymmetry. However, the user can adaptively adjust the weights in real-time to search for the most suitable weight(s) and solution(s).

### D.8 PROBLEM WITH MORE OBJECTIVES

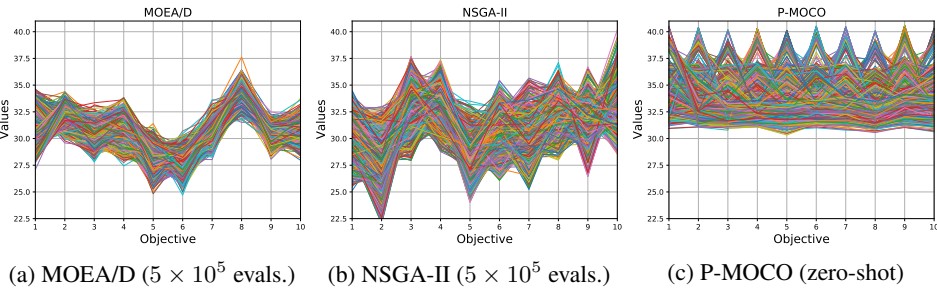

(a) MOEA/D ($5 \times 10^5$ evals.)    (b) NSGA-II ($5 \times 10^5$ evals.)    (c) P-MOCO (zero-shot)

Figure 12: The value path plots for the 10-objective MOKP100 obtained by MOEA/D, NSGA-II and our proposed P-MOCO. In the plot, each line (value path) represents a solution's 10 objective values with its specific preference. In MOKP, we want to maximize the values for all objectives. Our proposed method has significantly better overall performance.

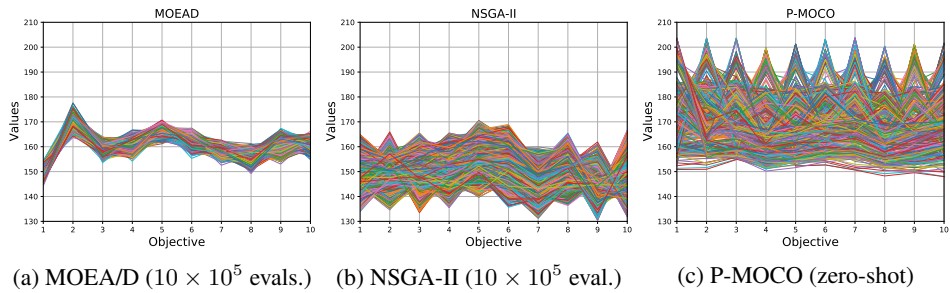

(a) MOEA/D ($10 \times 10^5$ evals.)    (b) NSGA-II ($10 \times 10^5$ eval.)    (c) P-MOCO (zero-shot)

Figure 13: The value path plots for the 10-objective MOKP with 500 items obtained by MOEA/D, NSGA-II and our proposed P-MOCO. Our model is trained on 10-objective MOKP100.

Finally, we test the performance of our proposed method on the 10-objective knapsack problems. We train a new model for the 10 objective MOKP with 100 items with uniform 10-dimension preferences. The obtained value path plots on the 10-objective MOKP100 are shown in Figure 12. For problems with more objectives, we need a large number of solutions to approximate the Pareto set. Training a large number of neural network models would have a huge computational and storage overhead, which is also not desirable in practice. Therefore, we do not compare with the AM-MOCO and MOA-DRL methods on this problem.

For inference, to approximate the Pareto set, we use a set of 715 fixed preferences following the weight assignment approach from (Das & Dennis, 1998) (with $m = 10, p = 4$, hence $n = C_4^{10+4-1} = 715$). The model generates different trade-off solution for each preference, so there are 715 different value paths (lines) on each plot. In MOKP, we want to maximize the values for all objectives under the capacity limitation. A set of good approximate solutions should have relatively high overall values. According to the results, our proposed method has the best performance. We also test the performance of our method on a larger problem with 500 items. The results shown in Figure 13 confirm that our trained model generalizes well to problems with a larger size.

