# OpenReview forum: "Pareto Set Learning for Neural Multi-Objective Combinatorial Optimization"
_ICLR.cc/2022/Conference — ICLR 2022 Poster_

### Official Review · Reviewer_CKHU · 2021-10-27

**Correctness:** 3
**Technical Novelty And Significance:** 3
**Empirical Novelty And Significance:** 4
**Recommendation:** 6
**Confidence:** 4

**Main Review:**

### Strengths
1. Multi-objective combinatorial optimization is a family of important problems but is really challenging to solve by traditional methods. The efforts of introducing deep reinforcement learning to this direction are appealing and novel. This paper may inspire more ML researchers in this interesting direction.
1. The experiment results seem detailed and sound. The proposed neural network MOCO method outperforms traditional approaches and other single-objective deep learning baselines.

### Weaknesses
1. The theoretical part of this paper (Section 6 and appendix A) does not seem sound to me. The assumption that the model can generate $\epsilon$-dominate solutions for any preference seems to be non-trivial for models with small enough $\epsilon$.

### Other comments
1. The hypervolume (HV) metric should be discussed in the main paper instead of in the supplementary material to ensure the main paper is self-contained.

**Summary Of The Paper:**

This paper presents a learning approach for multi-objective combinatorial optimization (MOCO), which is a challenging problem but not well-studied by previous machine learning researchers. The proposed model is capable of predicting approximate Pareto optimal solutions from various preferences by a single model, via attention networks, and by a so-called "hypernetwork". Experiment result on the multi-objective versions of TSP, VRP, and KP shows the effectiveness of the proposed approach.

**Summary Of The Review:**

In general, this paper is novel and sound, and the experiment results are convincing. I am suggesting a borderline accept of this paper, and it will be better if the authors could update the theoretical part of this paper and rearrange some materials to make this paper self-contained.

---

> ### Author Response · Authors · 2021-11-18
> **Response to Reviewer CKHU**
>
> Thank you for your time and effort in reviewing our work, and the valuable comments. We are glad to know you find our work is appealing and novel, has detailed and sound experiment results on challenging MOCO problems, and may inspire more ML researchers in this interesting direction. We address your concerns as follows.
>
> > **1. Theoretical Contribution**
>
> **Concerns on the Strong Assumption:** As you (and Reviewer mTSr) correctly pointed out, the theoretical approximation guarantee of our proposed method heavily depends on its ability to (approximately) solve each weighted single objective subproblem as discussed in Section 6 and Appendix A. Due to the NP-harness, it is indeed non-trivial to give a convergence guarantee to generate $\varepsilon$-dominate solutions for any preference with a small enough $\varepsilon$. This limitation also applies for other end-to-end learning-based (e.g, neural combinatorial optimization) and heuristic-based methods.
>
> **Learning-based Method for Exact Solution:** We are aware that some efforts have been made to combine the learning-based method with dynamic programming to achieve asymptotically optimal solution solution for specific single-objective problem in recent works [1,2]. These methods provide a controllable trade-off between the solution quality and the computational cost for solving NP-hard problems. However, their generalization to the multi-objective problem is not straightforward, since the scalarized subproblem for each preference is not necessary the same as its single-objective counterpart. For example, a Tchebycheff scalarized MOTSP is not a single-objective TSP as discussed at the end of Subsection 3.2. In addition, according to Bengio et al.(2020) [3], these methods belong to the class of learning alongside the algorithms, while our proposed approach is learning to directly predict the solutions (neural combinatorial optimization). Therefore, the idea for learning enhanced multi-objective combinatorial algorithm could be an important research topic in future, but out of the scope for the current work.
>
> **Our Theoretical Contribution:** The key contribution of this work is to approximate the whole (potentially exponentially large) Pareto set, and our approximation analysis is to provide a guarantee for the ideal case. We agree with the reviewer that the current theoretical contribution is limited because it depends on a strong assumption to (approximately) solve each NP-hard subproblem. However, it also provides valuable insights and a better understanding for our proposed method, which could be useful to the reader and inspire follow-up works.
>
> We have added Appendix A.3 to clearly discuss the limitation of our theoretical analysis and potential future work.
>
> [1] Cappart, Quentin, Thierry Moisan, Louis-Martin Rousseau, Isabeau Prémont-Schwarz, and Andre A. Cire. Combining Reinforcement Learning and Constraint Programming for Combinatorial Optimization. AAAI 2021.
>
> [2] Kool, Wouter, Herke van Hoof, Joaquim Gromicho, and Max Welling. Deep Policy Dynamic Programming for Vehicle Routing Problems. arXiv:2102.11756, 2021.
>
> [3] Bengio, Yoshua, Andrea Lodi, and Antoine Prouvost. Machine learning for combinatorial optimiza-tion: a methodological tour d’horizon.European Journal of Operational Research, 2020.
>
>
> > **2. Hypervolume (HV) Metric**
>
> Thank you for pointing this out. We have moved the key definition of the hypervolume metrics back to the main paper to make it self-contained, and also provided a more detailed discussion in the appendix due to the page limit (no extra page is allowed for this year's response).
>
> Please let us know if you have any other concerns. We are happy to take your further suggestion to improve our work.

---

> > ### Comment · Reviewer_CKHU · 2021-11-28
> > **Update**
> >
> > Thank the authors for the feedback, and here is a remark to update your camera-ready version: I am personally suggesting removing the theoretical part from the main paper because the assumption is relatively strong, and with all your assumptions and definitions, it is quite straightforward to achieve the theoretical conclusion in your paper.

---

> > > ### Author Response · Authors · 2021-11-28
> > > **Response to the Update**
> > >
> > > Thank you for your suggestion. We will move the theoretical part to the appendix and provide more discussions on the proposed algorithm in the camera-ready version (the manuscript cannot be edited in this phase).

---

### Official Review · Reviewer_mTSr · 2021-11-02

**Correctness:** 4
**Technical Novelty And Significance:** 4
**Empirical Novelty And Significance:** 3
**Recommendation:** 8
**Confidence:** 3

**Main Review:**

In my opinion, the authors provide an interesting and novel approach to multi-objecive optimization problems. Even representing the potentially exponentially large pareto set is a challenge, and is interesting that this can be done via an ML model. A concern could be that in order to achieve a good approximation of the pareto set, it seems that learning the single objective problem well is required (which often is a difficult problem in itself). Would it be possible to solve the single-objective variants with a traditional combinatorial algorithm? The theoretical contribution of this submission is limited.

The approach is evaluated for three very relevant problems: the multi-objective traveling salesperson problem, the multi-objective capacitated vehicle routing problem and the capacitated knapsack problem.

A minor concern is that I found the tables with the experimental results hard to read. Could you please add a short explanation for the meaning of the table/column headers?

**Summary Of The Paper:**

This submission treats multi-objective combinatorial optimization problems and aims to approximate the pareto set. The idea is to build a single ML model that represents the pareto set by providing a pareto set solution for any desired trade-off. The model is build using reinforcement learning and may either be used to find singular solutions with fixed trade-off, or to approximate the pareto set with uniform samples. The authors prove that the pareto set is approximated well if individual trade-offs are approximated well.

**Summary Of The Review:**

I feel this is an interesting contribution concerning a difficult and relevant problem, whose well-made practical part makes up for a more limited theoretical study.

---

> ### Author Response · Authors · 2021-11-18
> **Response to Reviewer mTSr**
>
> Thank you for your time and effort in reviewing our work, and the valuable comments. We are glad to know you find our work is interesting and novel, have strong practical performance on difficult and relevant problems, and give us a clear acceptance. We address your concerns as follows.
>
> > **1. Concern on Single Objective Subproblem and Theoretical Contribution**
>
> **Single Objective Subproblem:** As you (and Reviewer CKHU) correctly point out, the theoretical approximation guarantee of our proposed method heavily depends on its ability to solve each weighted single objective subproblem as discussed in Section 6 and Appendix A. Due to the NP-harness, it is challenging to give a convergence guarantee to the single objective subproblem for both the learning-based methods and traditional combinatorial algorithm (e.g., LKH for TSP) even they can achieve promising solutions in practice.
>
> There is no straightforward way to directly use a traditional combinatorial algorithm (e.g., LKH) to solve the weighted single objective subproblem. The major concern here is that the weighted Tchebycheff scalarized subproblem (e.g., Scalarized MOTSP), which involves min-max optimization, does not have the same form as its single objective counterpart (e.g., single objective TSP). Therefore, the traditional solvers for single objective problem cannot be used to solve the Tchebycheff scalarized subproblem. The simple weighted-sum MOTSP is a single objective TSP, but the weight-sum approach can only find the convex hull for a multi-objective optimization problem. We provide a detailed discussion on this issue at the end of Subsection 3.2.
>
> **Learning Enhanced Traditional Algorithm:** We are aware that some efforts have been made to combine the learning-based method with dynamic programming to achieve the asymptotically optimal solution for the specific single-objective problem in recent works [1,2]. These methods provide a controllable trade-off between the solution quality and the computational cost for solving NP-hard problems. However, due to the above discussion, their generalization to the multi-objective problem is not straightforward. In addition, according to Bengio et al.(2020) [3], these methods belong to the class of learning alongside the algorithms, while our proposed approach is learning to directly predict the solutions (neural combinatorial optimization). Therefore, the idea for learning enhanced multi-objective combinatorial algorithms could be an important research topic in the future, but out of the scope for the current work.
>
> **Our Theoretical Contribution:** The key contribution of this work is to approximate the whole (potentially exponentially large) Pareto set, and our approximation analysis is to provide a guarantee for the ideal case. We agree with the reviewer that the current theoretical contribution is limited because it depends on a strong assumption to (approximately) solve each NP-hard subproblem. However, it also provides valuable insights and a better understanding of our proposed method, which could be useful to the reader and inspire follow-up works.
>
> We have added Appendix A.3 to clearly discuss the limitation of our theoretical analysis and potential future work.
>
> [1] Cappart, Quentin, Thierry Moisan, Louis-Martin Rousseau, Isabeau Prémont-Schwarz, and Andre A. Cire. Combining Reinforcement Learning and Constraint Programming for Combinatorial Optimization. AAAI 2021.
>
> [2] Kool, Wouter, Herke van Hoof, Joaquim Gromicho, and Max Welling. Deep Policy Dynamic Programming for Vehicle Routing Problems. arXiv:2102.11756, 2021.
>
> [3] Bengio, Yoshua, Andrea Lodi, and Antoine Prouvost. Machine learning for combinatorial optimization: a methodological tour d’horizon.European Journal of Operational Research, 2020.
>
> > **2. Tables**
>
> Thank you for pointing this out. In the table, the meanings for each column header are:
>
> - **Instance Name (e.g., MOTSP20)** indicates the name of the problem (MOTSP) and the size (e.g., 20 cities);
> - **HV** stands for the hypervolume metric;
> - **Gap** is the ratio of hypervolume difference with respect to our method $\frac{\text{HV}{\text{ours}} - \text{HV}{\text{other algs}}}{\text{HV}_{\text{ours}}}$;
> - **Time** is each method's running time for solving 200 random test instances;
>
> We have moved part of the definition for hypervolume metric from the appendix to the main paper, and also added a short explanation for the column headers in the revised paper.
>
> Thank you again for all the valuable comments and suggestions. Please let us know if you have any other concerns. We are happy to take your further suggestion to improve our work.

---

### Official Review · Reviewer_7hop · 2021-11-03

**Correctness:** 4
**Technical Novelty And Significance:** 3
**Empirical Novelty And Significance:** 3
**Recommendation:** 6
**Confidence:** 2

**Main Review:**

The paper is well-written and presents an interesting approach to solving MOCO problems. The structure is appropriate, there is a very good review of related works, clear problem formulation, a good description of the method, experiments, and results. There are also extensive supplementary materials. The introduced method is novel, might be significant, and the quality of this article seems to be on-par with other papers applying ML techniques to solve TSP published at top-tier conferences (which are also cited in this paper). I don't see significant weaknesses. There are some minor typos (e.g., "a exceptionally" -> "an exceptionally", p. 4), so I recommend revising the paper before the final publication, but from the methodological point of view, the paper seems to be good enough to be accepted.

**Summary Of The Paper:**

The paper proposes a novel preference-conditioned method to approximate the whole Pareto front for Multi-Objective Combinatorial Optimization (MOCO) problems with a single model. According to the authors, this method provides extra flexibility for decision-makers to directly obtain arbitrary trade-off solutions without any extra search, which is a more principled way to deal with MOCO.

**Summary Of The Review:**

The paper is well-written and presents an interesting approach to solving MOCO problems. The structure is appropriate, there is a very good review of related works, clear problem formulation, a good description of the method, experiments, and results. There are also extensive supplementary materials. The introduced method is novel, might be significant, and the quality of this article seems to be on-par with other papers applying ML techniques to solve TSP published at top-tier conferences (which are also cited in this paper). I don't see significant weaknesses. There are some minor typos (e.g., "a exceptionally" -> "an exceptionally", p. 4), so I recommend revising the paper before the final publication, but from the methodological point of view, the paper seems to be good enough to be accepted.

---

> ### Author Response · Authors · 2021-11-18
> **Response to Reviewer 7hop**
>
> Thank you for your time and effort in reviewing our work. We are glad to know you find our work is novel, presents an interesting approach, well-written with high quality, and is good enough to be accepted.
>
> > **1. Minor Typos**
>
> Thank you for pointing this out. We have carefully revised the paper to correct the typos and potential grammar issues, and will continually proofread the paper before the final publication.
>
> > **2. Better Exposition and More Discussions**
>
> As suggested by other reviewers, we have revised and rearranged some parts of our work for better exposition. We have also provided more discussion on our proposed methods in both the main paper and the Appendix. Please see the response summary, our responses to other reviewers and the revised manuscript.
>
> Please let us know if you have any other concerns. We are happy to take your further suggestion to improve our work.

---

### Official Review · Reviewer_tRFG · 2021-11-04

**Correctness:** 3
**Technical Novelty And Significance:** 3
**Empirical Novelty And Significance:** 3
**Recommendation:** 6
**Confidence:** 3

**Main Review:**

Strengths

1. The proposed approach is technically sound and exhibits very competitive performance.
2. The empirical evaluation is thorough as it considers multiple test problems and a broad range of benchmark methods.

Weaknesses

1. The description of the proposed approach is hard to follow. In particular, I am having a hard time understanding how the decoder is defined when the problem instance is not a graph. In general, I would recommend the authors explain their approach in a general fashion by introducing appropriate notation and then explaining it in the context of a particular problem.

2. The use of the term "preference" is confusing. The title and abstract suggest that the proposed approach allows the incorporation of user preferences to focus the search on especific regions of the Pareto front, but that does not seem to be the case. I would suggest the authors to clarify this earlier (and even drop the term preference from their title).

3. Since the proposed method relies on scalarizations, it suffers from several well-known issues of this kind of approaches. In particular, scalarization-based approaches are prone to explore the Pareto front unevenly. The points generated in Figures 2 and 5 are well distributed accross the Pareto front, but I think occurs because the front in this particular case is fairly symmetric. I wonder how an analogous figure would look for a more irregular front. I would also like the authors to explain how the weights that give rise to the points in Figures 2 and 5 were chosen.

4. The novelty in this paper is limited, as it reuses several ideas from single-objective neural combinatorial optimization. The only novel (but key) idea is to make the decoder weight-dependent.

**Summary Of The Paper:**

This paper proposes an approach for neural multi-objective combinatorial optimization. This approach uses a preference-agnostic encoder along with a weight-dependent decoder to generate approximate Pareto optimal solutions for any arbitrary set of weights of a weighted Tchebyshev scalarization at virtually no additional cost. This is in contrast with existing approaches, which require a significant amount of computation for every new set of weights. Several numerical experiments are conducted, showing favorable results for the proposed method when compared with several other existing evolutionary and learning-based methods.

**Summary Of The Review:**

This paper proposes a novel approach for neural multi-objective combinatorial optimization. The novelty of the proposed approach is limited as it reuses several ideas from single-objective neural combinatorial optimization. However, it is technically sound and exhibits a very competitive performance in a broad range of problems. The description of the method is hard to follow but, in general, the paper is well-written. The use of the term "preference" is confusing, so I would recommend the authors explain more clearly that their approach does not incorporate preference information.

---

> ### Author Response · Authors · 2021-11-18
> **Response to Reviewer tRFG [2/2]**
>
> > **3. Weight Assignments and Unevenly Distributed Solutions**
>
> This is a valuable point. We answer these three closely related questions in inverse order.
>
> **Weight Assignments:** We use the structured weight assignment approach from Das and Dennis (1998) [1] to give the weights for Figure 2 and Figure 5. This method can generate $n = C_p^{m + p -1}$ evenly distributed weights with an identical distance to their nearest neighbor on the unit simplex (e.g., $\sum_{i=1}^{m} \lambda_i = 1$ with $\lambda_i \geq 0,\forall i$), where $m$ is the number of objectives and $p$ is a parameter to control the number of weights.
>
> For the three objective TSP problems ($m = 3$), we assign $p = 13, 44$ and $140$ to generate $n = 105, 1035$ and $10011$ uniform weights respectively. We have expanded Subsection D.3 in the Appendix to illustrate how we choose these weights, and add a pointer to this part in the description of Figure 2.
>
> **Asymmetric Pareto front:** As you correctly pointed out, a drawback of the scalarization-based approach is that it cannot evenly explore the asymmetric Pareto front with a set of uniform weights. In the original manuscript, we have provided a preliminary study on this issue with MOCVRP, of which the Pareto front is asymmetric in Appendix D.4. Following your suggestion, we have now added results for MOTSP with asymmetric Pareto front in Appendix D.5.
>
>
> Thanks to the ability to generate arbitrary trade-off solutions, our approach allows the users to adaptively adjust the weight within their preferred region on the inference time, or directly generate a dense approximation. This flexibility can partially address the unevenly distributed issues caused by a set of fixed weights in the traditional scalarization-based approach. We have added new experiments on preference-based inference in Appendix D.6.
>
> **Adaptive Weight Assignments:** If we know the approximate range of different objectives, we can first normalize them into [0,1] to encourage a more symmetric Pareto front. For the problem with a truly irregular Pareto front, it is also possible to adaptively adjust the given weights to make them evenly explore the Pareto front during the learning/searching process. One potential direction could be to consider the connection between scalarization and hypervolume maximization as in Golovin and Zhang (2020) [2]. We believe this could be an important research topic for the learning-based scalarization approach in future work.
>
>
> [1] Das, I. and Dennis, J.E., 1998. Normal-boundary intersection: A new method for generating the Pareto surface in nonlinear multicriteria optimization problems. SIAM journal on optimization, 8(3), pp.631-657.
>
> [2] Golovin, D. and Zhang, Q., 2020. Random hypervolume scalarizations for provable multi-objective black box optimization. NeurIPS 2020.
>
> > **4. Simple yet Key Idea on Weight-Dependent Decoder**
>
> We are glad to know you appreciate our simple yet key idea on the weight-dependent decoder to achieve the technically sound and very competitive method for a broad range of MOCO problems.
>
> For the concern on novelty, we agree with you that our method combines and extends several ideas from single-objective neural combinatorial optimization. We also want to emphasize our novel and key contributions on neural MOCO, some of which have already been recognized by the reviewer:
>
>
> - We propose a novel method to approximate the whole Pareto set  {$x(\boldsymbol{\lambda})|\sum_{i=1}^{m}\lambda_i = 1$} with infinite preferences via a single model, rather than a fixed small subset of solutions {$x_1, x_2,\cdots,x_p$} (e.g., $p =  101$) in the current heuristic-based and learning-based methods. We believe it is a more principle way to deal with multi-objective combinatorial optimization.
> - Our method's model efficiency is crucial to truly generalize the learning-based method to solve multi-objective combinatorial optimization problems, for example, generating more than $10,000$ different solutions to approximate the Pareto front for 3-objective MOTSP.
> - We believe simplicity is also an advantage for our method. Due to the design choice of minimal essential change (e.g., the weight-dependent decoder), our method is agnostic to the encoder design, and can also enjoy the current improvements that were originally proposed for the single objective NCO.
>
> We believe our method is an important extension to truly generalize neural combinatorial optimization to solve a broad range of challenging MOCO problems, which could inspire more future work in the community.
>
> Thank you again for all the valuable comments and suggestions. Please let us know if you have any other concerns. We are happy to take your further suggestion to improve our work.

---

> ### Author Response · Authors · 2021-11-18
> **Response to Reviewer tRFG [1/2]**
>
> Thank you for your time and effort in reviewing our work, and the valuable suggestions. We are glad to know you find our proposed approach is technically sound, exhibits very competitive performance, and has a thorough empirical evaluation. We address your concern as follows.
>
> > **1. Detailed Description of the Proposed Approach**
>
> Thank you for this suggestion. we have extended the decoder subsection in Section 4.2 to give a more detailed explanation of the decoder model. We take a MOTSP instance as an example to illustrate how the decoder sequentially selects the cities to construct a valid tour.
>
> For a MOTSP instance, the main approach works as follows: 1) the encoder generates the embeddings for all cities; 2) the decoder calculates the probabilities for selecting each city based on the embeddings; and 3) the decoder select the next city to visit in sequence (repeated 2) and 3)). Our proposed model only needs a single forward pass through the dense encoder (around 90% parameters). The weight-dependent decoder can generate different solutions (tours) of different trade-offs with the same set of city embeddings.
>
>
> > **2. The Term "Preference"**
>
> Thank you for pointing this out. In this work, we focus on approximating the whole Pareto set by a single model, which allows users to use any trade-off preferences (via the weights for Tchebycheff scalarization) at the inference time to obtain their preferred solutions. The term "preference" is used interchangeably with the term "weight" in scalarization.
>
> As you correctly pointed out, the term "preference-conditioned" in the title might lead to misunderstanding with the preference-based search. We have changed the title to "Pareto Set Learning for Neural Multi-Objective Combinatorial Optimization". The new title emphasizes the key idea is to use a learning-based algorithm to approximate the whole Pareto set. We have also modified the description in Abstract and Introduction, and added a brief clarification in Section 3.3 to clarify this point.
>
> We did not discuss the incorporation of user preferences in the training process (e.g., to focus the learning on specific regions). But, indeed, our proposed method does have an efficient way to incorporate user preference, via the weight distribution assignment for training and/or active search (Equation 7). In the current work, we use a simple uniform distribution $\Lambda$ for preference sampling, which can be easily replaced by a user-defined and biased distribution with respect to their preferences. In the extreme case, if the user only cares about a specific fixed trade-off among the objectives, our proposed approach will reduce to the single-objective neural combinatorial optimization.

---

### Author Response · Authors · 2021-11-18
**Response Summary**

We thank all reviewers for their constructive and valuable comments. Following their suggestion, we have made the following revision, which is also highlighted in **orange** in the revised paper:

- **Better Emphasis on Our Key Idea:** We have changed the title to "Pareto Set Learning for Neural Multi-Objective Combinatorial Optimization" which can better emphasize our key learning-based approach to approximate the whole Pareto set, and avoid misunderstanding. We have also revised the Abstract and Introduction accordingly.
- **Detailed Description for Preference-based Decoder:** We have revised Subsection 4.2 to better define the preference-based decoder, and make it clear how the decoder generates a valid solution.
- **Self-Contained Experimental Setting:** We have moved the definition of hypervolume metrics back to the main paper, and also added a short explanation for the column headers in the tables of experimental results.
- **More Experiments:** We have revised Appendix D.3 to explain the weight assignment method in the experiment, and provided more experimental results on MOTSP with asymmetric Pareto front in Appendix D.5, and preference-based inference in Appendix D.6.
- **Limitation for Theoretical Analysis:** We have added Appendix A.3 to clearly discuss the limitation of our theoretical analysis and potential future work.
- **Rearrangement of the Active Adaption Subsection:** Due to the page limit (no extra page is allowed this year), we have moved part of Subsection 5.3 on active adaption to the Appendix.

Details point-to-point responses can be found below to each reviewer. We are also glad to continually improve our work to address any further concern.

Paper2479 Authors

---

### Decision · Program_Chairs · 2022-01-20

**Decision:**

Accept (Poster)

**Comment:**

This paper develops a ``preference-conditioned” approach to approximate the Pareto frontier for Multi-Objective Combinatorial Optimization (MOCO) problems with a single model (thus dealing with the thorny problem that there can be exponentially-many Pareto-optimal solutions). It appears to provide  flexibility for users to obtain various preferred tradeoffs between the objectives without extra search. The basic idea is to use end-to-end RL to train the single model for all different preferences simultaneously.

The technical soundness and practical performance are strong. This work's approximation guarantee depends on the ability to approximately solve several (weighted) single-objective problems. This may be challenging due to the NP-hardness of the latter. However, this limitation seems to also apply to other end-to-end learning-based approaches.

One area where the novelty is somewhat limited is that the paper borrows some number of ideas from neural single-objective optimization. The contribution overall seems noteworthy for hard multi-objective problems.